# Epidermal galactose spurs chytrid virulence and predicts amphibian colonization

Yu Wang[1]✉, Elin Verbrugghe [1], Leander Meuris [2], Koen Chiers[3], Moira Kelly[1], Diederik Strubbe[4], Nico Callewaert[2], Frank Pasmans[1,5] & An Martel [1,5]✉

The chytrid fungal pathogens *Batrachochytrium dendrobatidis* and *Batrachochytrium salamandrivorans* cause the skin disease chytridiomycosis in amphibians, which is driving a substantial proportion of an entire vertebrate class to extinction. Mitigation of its impact is largely unsuccessful and requires a thorough understanding of the mechanisms underpinning the disease ecology. By identifying skin factors that mediate key events during the early interaction with *B. salamandrivorans* zoospores, we discovered a marker for host colonization. Amphibian skin associated beta-galactose mediated fungal chemotaxis and adhesion to the skin and initiated a virulent fungal response. Fungal colonization correlated with the skin glycosylation pattern, with cutaneous galactose content effectively predicting variation in host susceptibility to fungal colonization between amphibian species. Ontogenetic galactose patterns correlated with low level and asymptomatic infections in salamander larvae that were carried over through metamorphosis, resulting in juvenile mortality. Pronounced variation of galactose content within some, but not all species, may promote the selection for more colonization resistant host lineages, opening new avenues for disease mitigation.

[1] Wildlife Health Ghent, Department of Pathology, Bacteriology and Poultry Diseases, Faculty of Veterinary Medicine, Ghent University, Merelbeke, Belgium. [2] Center for Medical Biotechnology, Department of Biochemistry and Microbiology, VIB-Ghent University, Zwijnaarde, Belgium. [3] Department of Pathology, Bacteriology and Poultry Diseases, Faculty of Veterinary Medicine, Ghent University, Merelbeke, Belgium. [4] Terrestrial Ecology Unit, Department of Biology, Faculty of Sciences, Ghent University, Ghent, Belgium. [5]These authors contributed equally: Frank Pasmans, An Martel. ✉email: Yu.Wang@ugent.be; An.Martel@ugent.be

  1

Mitigation of infectious diseases has become a key challenge in curbing biodiversity loss. One of the wildlife diseases contributing to Earth's sixth mass extinction is the lethal skin disease chytridiomycosis. This fungal disease is linked to the extinctions or declines of hundreds of amphibian species worldwide[1,2]. Efforts to contain the impact of the disease, including host removal, the use of chemical disinfectants, probiotics, and habitat alteration, have thus far had limited success[3–6], and such actions can themselves be controversial, given the dire state of amphibian populations and their often vulnerable habitats. Designing more targeted and sustainable mitigation strategies, for example, vaccination or selective breeding of resistant host lineages, requires a thorough understanding of host–pathogen–environment interactions.

Chytridiomycosis is caused by the chytrid fungi *Batrachochytrium dendrobatidis*[7] and *B. salamandrivorans*[8]. The latter was recently discovered from a collapsing fire salamander (*Salamandra salamandra*) population in the Netherlands[8] and is causing mass mortality events in wild salamander populations across Europe[6,9,10]. In susceptible animals, *B. salamandrivorans* causes epidermal necrosis[8], resulting in loss of the epidermal barrier and subsequent overgrowth and invasion by opportunistic bacteria, bacterial septicemia, and death[11].

Infection of the amphibian skin by the motile fungal spores[12,13] is a complex and poorly understood process that requires recognition of, attraction to, and subsequent attachment to the outer layers of the skin, in order to be able to invade the skin surface. To successfully invade the skin the fungus needs to overcome the physical (mucus and stratum corneum), chemical (antimicrobial peptides and toxins), cellular (immune cells), and microbiological barriers of the epidermal layer[14]. Little is known about the molecules that participate in the early interactions between pathogen and host, but they likely include polypeptides (proteins) or polycarbohydrates (carbohydrates), since fungal ligand-host receptor binding is mediated mainly by either protein−protein or protein−carbohydrate interactions[15,16]. The first matrix encountered is the epidermal mucosome, comprising host mucus and other host secreted and microbiome-derived compounds. This saccharide-rich environment contains oligosaccharides that are known to attract *B. dendrobatidis* zoospores[12,17]. Following attachment, the *B. salamandrivorans* arsenal of proteases is thought to play a key role in subsequent cell invasion[18].

The extent of epidermal infection correlates with the severity of the disease, which varies strongly between, and even within, amphibian host species[6,12,19,20]. The outcome of infection depends on complex host, pathogen, and environmental interactions and can vary from the absence of clinical signs to rapid death[19]. Predicting host susceptibility is crucial for risk assessments and the development of mitigation action plans and is currently only possible using invasive infection experiments[6].

In this work, we unravelled the early interactions of *B. salamandrivorans* with its amphibian host and investigate the potential use of skin galactose as a biomarker for fungal colonization. Using in vitro assays we show that beta-galactose mediates fungal attraction and adhesion to the amphibian skin. Beta galactose selectively upregulates virulence-associated fungal genes and increases protease activity in zoospores, suggesting the initiation of a virulent fungal response. Histochemistry of the skin of 9 urodele and 5 anuran species and of different life stages of fire salamanders (*Salamandra salamandra*) demonstrates a marked variation of the cutaneous glycosylation pattern of amphibian skin between species and life stages, notably of galactose, which correlates with susceptibility to chytrid colonization. A similar correlation between the proportion of galactose in the total carbohydrate fraction of skin washes of 17 urodele and 4 anuran species and chytrid colonization corroborates the use of galactose as a biomarker for susceptibility to *B. salamandrivorans* colonization. While *B. salamandrivorans* infections in salamander larvae are asymptomatic, infections may be carried over to metamorphosis, resulting in lethal infections of juveniles. Significant variation of cutaneous galactose content in two of three urodele species examined, further suggests the possibility of selection for increased colonization resistant lineages in some urodele species.

## Results and discussion

**Galactose mediates *B. salamandrivorans* attraction and adhesion to salamander skin.** We first identified key factors in the attraction and attachment of *B. salamandrivorans* to the amphibian skin, representing the first steps in the pathogenesis of *B. salamandrivorans* infection. We investigated if *B. salamandrivorans* spores bind to the carbohydrate or protein fraction of the amphibian skin. A water-soluble lysate was prepared from sloughed skin of fire salamanders (*Salamandra salamandra*) and this crude preparation was used to compare the binding ability of *B. salamandrivorans* spores to precipitated skin proteins, precipitated skin proteins lacking carbohydrates (through enzymatic deglycosylation), and supernatant of the latter, containing the enzymatically removed carbohydrate fraction. *B. salamandrivorans* zoospore binding was quantified by counting the attached zoospores in a binding assay. While binding of spores was comparable between the crude preparation, the precipitated protein fraction and the carbohydrate fraction, binding to the deglycosylated proteins was significantly reduced, compared to the crude preparation ($p = 0.0002$), precipitated protein fraction ($p = 0.0018$), and carbohydrate fraction ($p < 0.0001$) (Fig. 1a and Supplementary Tables 1, 2). *B. salamandrivorans* zoospores thus predominantly bind to carbohydrates of salamander skin.

To identify the carbohydrates involved in *B. salamandrivorans* binding, we then coated a series of carbohydrates on microtiter plates and quantified the number of attached zoospores. The fungal spores predominantly bound to lactose, N-Acetylgalactosamine (GalNAc), and mannose, but not to N-Acetylglucosamine (GlcNAc) (Fig. 1b and Supplementary Tables 3, 4), identifying galactose and mannose as *B. salamandrivorans* binding sites on salamander skin. In addition, a capillary tube chemotaxis assay demonstrated movement of *B. salamandrivorans* zoospores towards carbohydrates, with a high affinity towards galactose (Fig. 1c and Supplementary Table 5). This is in contrast with *B. dendrobatidis*, which is attracted by the tested carbohydrates (mannose, galactose, fucose, N-acetylglucosamine, N-acetylgalactosamine, and N-acetylneuraminic acid) without specific preference[12]. To corroborate these findings, we searched for evidence of carbohydrate-binding proteins in the *B. salamandrivorans* genome (AMFP13/01, NCBI database, Bioproject PRJNA311566), which yielded two ricin B lectins, two legume-like lectins and one concanavalin A (Con A)-like lectin (Supplementary Table 6). Ricin B lectin binds oligosaccharides containing either terminal beta-GalNAc or β-1,4-linked galactose residues[21]. Both legume-like lectins were identified as mannose-specific lectins[22,23], and the Con A-like lectin binds specifically to α-D-mannosyl and α-D-glucosyl residues[24]. The expression of a ricin B lectin gene (BSLG_00833) and the Con A-like lectin gene (BSLG_02674) upon *B. salamandrivorans* exposure to salamander skin was demonstrated previously[18]. These results suggest that *B. salamandrivorans* expresses lectins to bind to galactose and mannose in the salamander skin.

**Contact with galactose spurs fungal virulence.** To better understand how these carbohydrates mediate the early stages of *B. salamandrivorans* infection, we performed RNA-seq analysis and

a

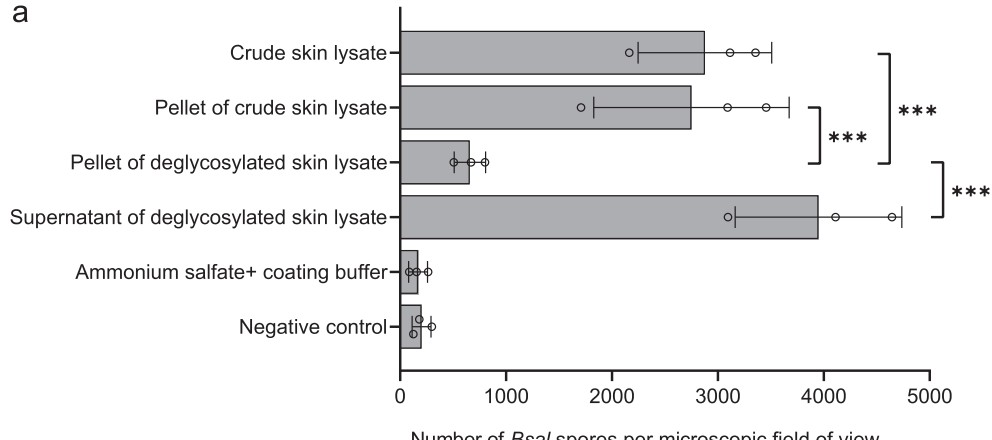

b

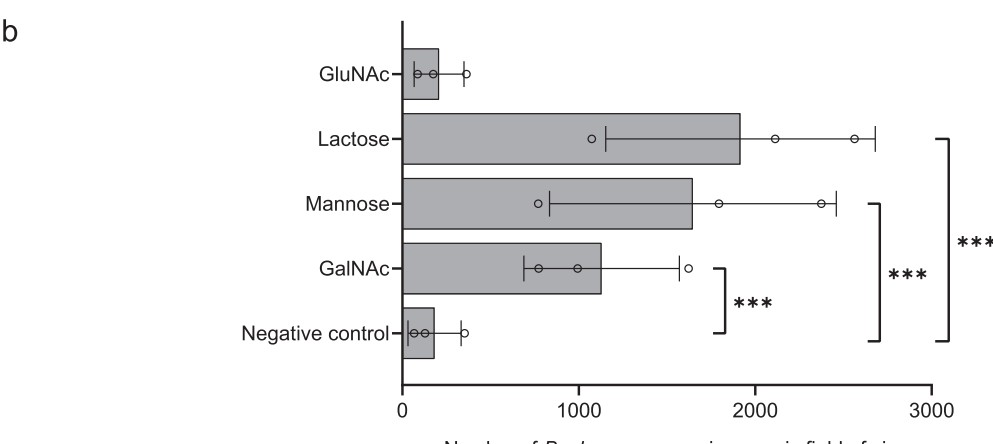

c

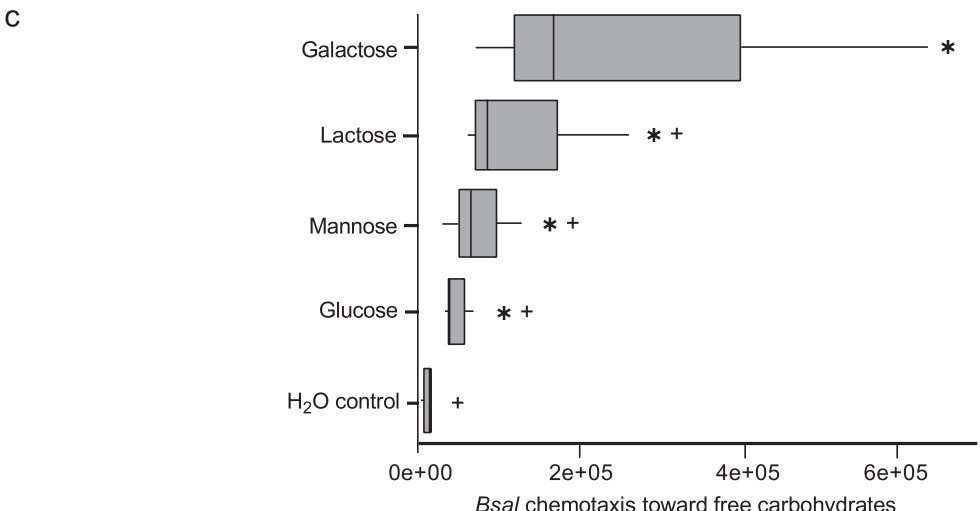

found a selective upregulation of putative *B. salamandrivorans* virulence factors during initial contact with carbohydrates and galactose in particular. RNA-seq analysis detected 25 uniquely expressed genes in galactose-treated zoospores (Fig. 2a). Among those, BSLG_05880 and BSLG_09248 were annotated as protein tyrosine kinase candidates, which are known to participate in signalling transduction pathways and regulate a series of essential cellular processes, such as cell growth, differentiation, and death[25]. In choanoflagellates, protein tyrosine kinases have been found to regulate cell proliferation[26] and react to the environmental nutrient

**Fig. 1 Galactose mediates *B. salamandrivorans* attraction and adhesion. a** *B. salamandrivorans* zoospores bound on microtiter plates coated with crude and enzymatically treated skin lysate of sloughed fire salamander skin. Pellet and supernatant fractions were obtained by protein precipitation. **b** *B. salamandrivorans* zoospores bound on microtiter plates coated with different carbohydrates. GlcNAc = N-Acetylglucosamine, GalNAc = N-Acetylgalactosamine. **a, b** Negative control = wells coated with coating buffer only. Three technical replicates were performed per biological replicate, with three biological replicates in total. Figures represent the mean and variation (standard deviation) of the biological replicates. Black circles represent biological replicates. The significance of the difference between the distributions is shown by *** $p < 0.001$, based on Tukey's multiple comparisons with the glht function in R package multcomp, comparing (two-sided) negative binomial models. **c** Chemotaxis of *B. salamandrivorans* toward free carbohydrates. The sugars D-glucose, D-mannose, lactose, and D-galactose were tested as attractants at a 0.1 M concentration, using a traditional capillary tube test. Water was used as a vehicle and control attractant. Genomic equivalents (GE) of *B. salamandrivorans* zoospores in the capillaries were quantified after 90 min using quantitative realtime PCR. At least three technical replicates were performed per biological replicate, with three biological replicates in total. Box-and-whisker plot presents the mean and variation in chemotaxis of the biological replicates; the median is shown as a vertical line inside the box, the first and third quartiles shown as the lower and upper edges of the box, respectively, and the minimum and maximum values shown as whiskers. An * indicates a significant difference compared to the water control, whereas + designates a significant difference compared to galactose; with adjusted *p*-values obtained via Tukey's multiple comparisons corrections. **a−c** Individual *P*-values are shown in Supplementary Tables 2, 4, and 5. Source data are provided as a Source Data file.

---

availability[27]. Protein tyrosine kinases have also been shown to play a crucial role during fungal infections, including attachment to host cells[28,29]. BSLG_08965, also specifically linked to galactose treatment, was annotated as belonging to the alpha/beta hydrolase family (Fig. 2a), which is one of the biggest groups of structurally related hydrolytic enzymes with diverse catalytic functions, such as hydrolases, lyases, or transporters of other proteins[30,31]. Alpha/beta hydrolase fold proteins have been reported to regulate the interactions between pathogenic bacteria, and were previously associated with the efficient infection and adaptation of hosts by *B. dendrobatidis*[32–34]. Gene groups uniquely expressed in mannose- and glucose-treated spores contained repair enzymes, proteins linked to transport, but also possible virulence candidates including protein kinase (mannose BSLG_00482 and glucose BSLG_08672) and copper/zinc superoxide dismutase (mannose BSLG_09793), which is known as a superoxide radical scavenger linked to fungal virulence[35]. Genes uniquely expressed in mannose were mainly associated with mitochondria, whereas the majority of the genes uniquely expressed in glucose seemed to be associated with the transport of cellular components and particularly proteins (e.g., secretory exocyst component, kinesin motor domain, transporters, nexin, clathrin, and exportin) (Supplementary Data 1).

Differential gene expression analysis showed a significant upregulation/downregulation of 1079/610, 1017/418, or 925/746 genes in, respectively, galactose, glucose, or mannose-treated zoospores, compared to the $H_2O$-treated zoospores (Fig. 2b). The fungalysin metallopeptidase (M36) family and the serine-type peptidase (peptidase S41) family are highly expanded in both *B. dendrobatidis* and *B. salamandrivorans*, compared to non-pathogenic chytrid fungi and are considered virulence factors, involved in the initial stages of zoospore colonization of amphibian skin and entry into host cells[18,36,37]. M36 metalloprotease candidates BSLG_08963 ($Log_2$ fold change vs $H_2O = 2.25$) and BSLG_09557 ($Log_2$ fold change vs $H_2O = 2.55$), together with the peptidase S41 candidate BSLG_06886 ($Log_2$ fold change vs $H_2O = 3.66$), showed an upregulation in galactose-treated zoospores, compared to $H_2O$-treated zoospores, but not in glucose- or mannose-treated zoospores (Fig. 2c). As a general response to carbohydrates, a significant upregulation ($Log_2$ fold change vs $H_2O \geq 2.0$) was observed in possible virulence genes including multiple protein kinase candidates (BSLG_01979, BSLG_09370, BSLG_07973, BSLG_05106, BSLG_09982, BSLG_01828, BSLG_02473 and BSLG_08449) and the peptidase family S41 candidate (BSLG_07398). Two-tailed Fisher's exact test for Gene Ontology (GO) terms in the genes significantly upregulated in response to carbohydrate exposure against the remaining gene set showed enrichment of kinase activity (GO:0016301) and transferase activity (GO:0016772) (FDR *p*-value < 0.05). Exposure to carbohydrates, and specifically

galactose, initiates a cascade of protein changes, including expression and upregulation of a number of virulence candidates.

This was translated to increased protease activity of zoospores. When exposed to galactose, protease activity is significantly higher than when mannose- ($p = 0.008$), $H_2O$- ($p = 0.036$) or protease inhibitor (PI)-treated ($p < 0.001$) (Fig. 3 and Supplementary Table 7). Increased protease activity corroborates the initiation of a virulent response of *B. salamandrivorans* upon contact with galactose.

**Life stage-dependent susceptibility to *B. salamandrivorans* colonization correlates with ontogenetic galactose patterns in the amphibian skin.** Marked differences in lectin binding patterns have been observed in amphibians[38], suggesting profound differences in carbohydrate patterns. Since *B. salamandrivorans* is attracted by and attaches to carbohydrates, the salamander skin carbohydrate content may predict the magnitude of host skin colonization. Ontogenetic carbohydrate patterns of different life stages of fire salamanders were compared to explain intraspecific differences of host susceptibility. While *B. salamandrivorans* infection in fire salamanders post metamorphosis is consistently lethal[8,19], larvae are not considered susceptible to chytridiomycosis[12]. *Ricinus communis* agglutinin (RCA) and concavalin A (Con A) histochemistry was used to score the presence of galactose and mannose/glucose in the skin, respectively. The presence of galactose but not mannose or glucose in the skin of fire salamander larvae markedly increases with age and climaxes towards metamorphosis (Fig. 4 and Supplementary Fig. 1). Post metamorphosis, these high galactose levels are maintained.

The correlation between galactose levels and susceptibility to *B. salamandrivorans* infection was studied by experimental exposure of different life stages of fire salamanders (early and late-stage larvae and metamorphs) to *B. salamandrivorans*. The infection prevalence and load was much higher in metamorphs than in larvae (Wilcoxon rank-sum test of *B. salamandrivorans* load in genomic equivalents $p = 0.0001$, z-score = $-3.885$ $n$(larvae) = 37, $n$(metamorphs) = 6; Fig. 4a). Disease signs (skin ulcerations) were noted in metamorphs only and not in the pre-metamorphic stages (Fig. 4b). Five out of ten inoculated late stage larvae carried over the infection through metamorphosis, resulting in lethal disease of juveniles. The infection load in the larvae correlated with the intensity of the galactose staining ($\beta = 2.81$, $p = 0.014$, $n = 43$, regression F-statistic $(2,40) = 5.40$ and *p*-value = 0.008; Fig. 4c−e). Infection load and disease course are thus correlated with ontogenetic galactose patterns in salamander skin.

**Skin galactose staining predicts *B. salamandrivorans* infection intensity and survival probability in amphibians.** We then

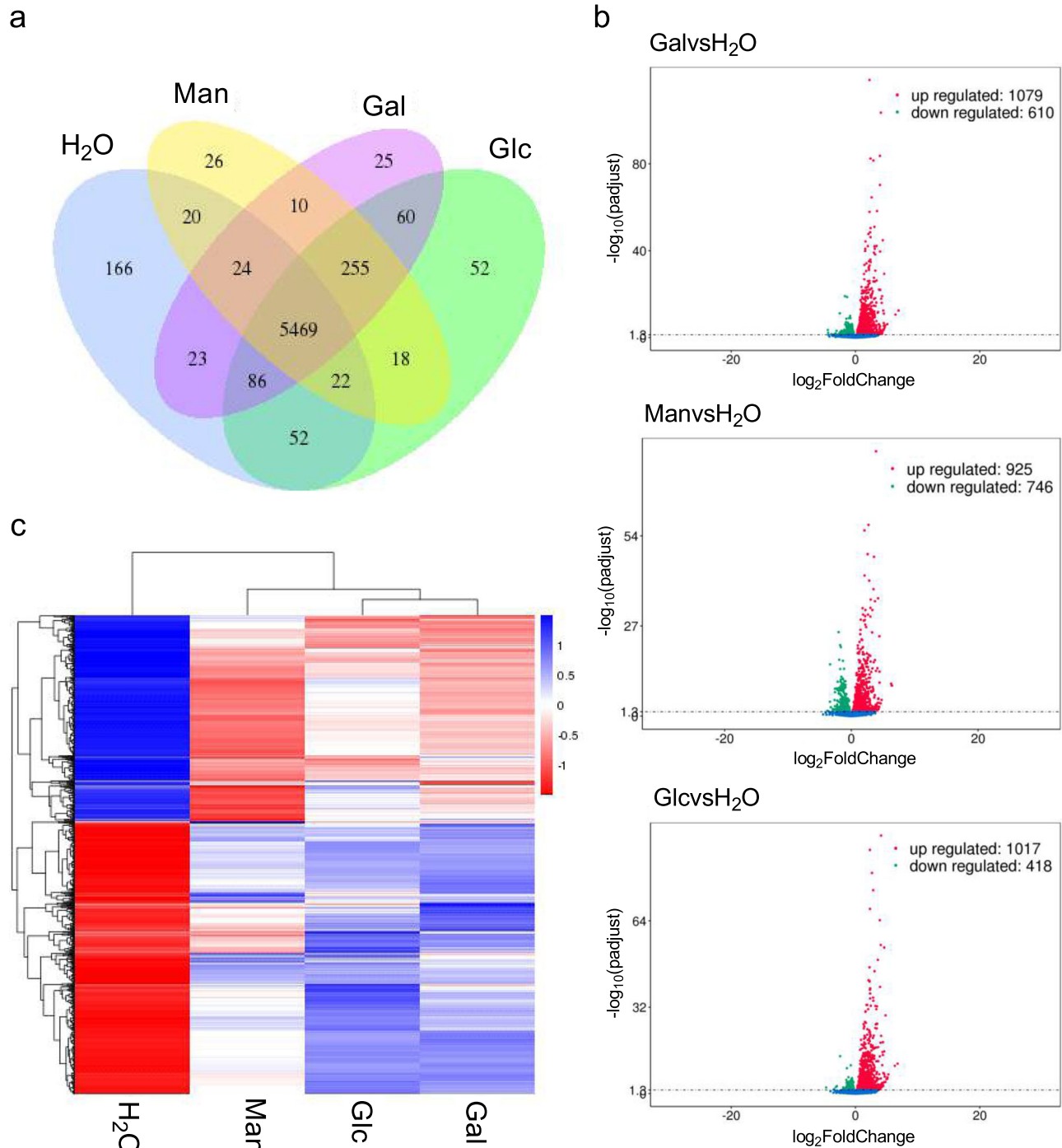

**Fig. 2 Gene expression in *B. salamandrivorans*. a** Co-expression Venn diagram presents the number of genes that are uniquely expressed within each group per sample, with the overlapping regions showing the number of genes that are co-expressed in two or more groups per samples. **b** Volcano plots of differentially expressed genes identified between the galactose, glucose, or mannose group and control group. The green dots denote downregulated gene expression, the red dots denote up-regulated gene expression, and the blue dots denote the gene expression without marked differences. **c** FPKM cluster analysis, clustered using the $\log_{10}$ (FPKM + 1) value. Blue denotes genes with high expression levels, and red denotes genes with low expression levels. The colour ranging from blue to red indicates $\log_{10}$ (FPKM + 1) value from large to small. Trees indicate hierarchical clustering between data sets (above) and genes (left of heatmap). Gal = galactose; Glc = glucose; Man = mannose. Source data are provided as a Source Data file.

compared the presence of epidermal galactose and susceptibility to *B. salamandrivorans* infection across fourteen amphibian species (nine urodelan species, five anuran species; Fig. 5). To exclude anatomical topology differences in carbohydrate pattern, we first compared the presence of galactose between different body sites (ventral and dorsal skin, toeclips, and tailclips) of the same animal in three species (alpine newts, fire salamanders, and palmate newts; Supplementary Table 8). The RCA staining was consistent in the skin of the different body parts. Hence, tailclips and toeclips were used for urodelan and anurans, respectively. We used average peak loads as a proxy for susceptibility.

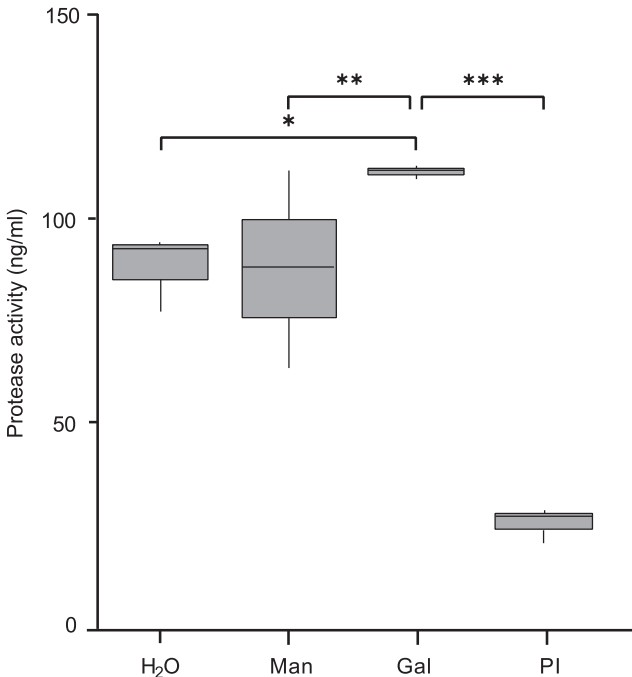

**Fig. 3 Protease activity detected in supernatants of *B. salamandrivorans* zoospores.** Supernatants were collected by centrifuging the *B. salamandrivorans* zoospore suspensions treated with 50 mM D-galactose (Gal), 50 mM D-mannose (Man), H₂O and protease inhibitor mix (PI). Three technical replicates were performed per biological replicate, with three biological replicates in total. In the box-and-whisker plots, the median is shown as a line inside the box, the first and third quartiles shown as the lower and upper edges of the box, respectively, and the minimum and maximum values shown as whiskers. The significance of difference between the means is shown by $*p = 0.036$, $**p = 0.008$, $***p < 0.001$; based on Tukey's multiple comparisons test comparing two-sided linear mixed models. Source data are provided as a Source Data file.

RCA histochemistry revealed a clear positive correlation between *B. salamandrivorans* infection peak loads and RCA staining intensity across the fourteen species included ($r_{pb} = 0.687$, $p = 0.007$; Fig. 5a and Supplementary Table 9). We confirmed this with a linear model using dummy variables for RCA scores and found that RCA scores can explain 87.2% of the variance in *B. salamandrivorans* infection peak loads ($R^2 = 0.872$; Supplementary Table 10). All seven species (five anuran species, two urodelan species) reported to be less susceptible to *B. salamandrivorans* infection showed a negative to weak RCA signal (Fig. 5 and Supplementary Table 9). These species were previously shown to develop low-level and asymptomatic infections[6,39]. The eight susceptible species that are extensively colonized by *B. salamandrivorans* upon exposure[6,19] showed marked galactose presence in their skin, represented by an intense RCA signal (Fig. 5 and Supplementary Table 9). We found that galactose content in the skin is a poor predictor of disease course after infection ($r_{pb} = 0.193$, $p = 0.509$, $R^2 = 0.564$; Fig. 5b and Supplementary Table 10). Indeed, extensive colonization needs not necessarily result in lethality. Rather, innate and acquired defence mechanisms and the environmental context are likely to determine the disease outcome[13]. Thus, RCA staining of the tail or toeclips can be useful as a predictor of susceptibility of a species for *B. salamandrivorans* colonization and, less reliably, for severity of disease progression. This information can inform mitigation strategies and action plans. We applied this to tailclips of the endangered Lanzai salamander (*Salamandra lanzai*), which showed intense RCA staining. Based on the dummy variable

regression models between RCA scores with infection peak loads and mortality rates (Supplementary Table 10), we predict that infection of *Salamandra lanzai* would result in high infection intensities and a high probability of lethal infections. This estimate is supported by its phylogenetic vicinity to closely related, and known susceptible species. We thus propose the galactose staining pattern in salamander skin to be a valuable predictor of infection intensity and, to a lesser extent, survival probability.

**Noninvasive sampling of amphibians for galactose levels as a biomarker for infection intensity.** While quantifying galactose in the skin may be a promising tool to predict amphibian susceptibility to *B. salamandrivorans* infection, noninvasive sampling is preferable over the collection of tissue samples. We therefore studied whether testing galactose levels in the amphibian skin mucosome[40] could be a viable alternative. Amphibian skin washes were collected by bathing animals from seventeen species of urodeles and four species of anurans in water for 1 h. The skin washes were subsequently examined for the concentration of oligosaccharides.

The proportion of galactose in the total carbohydrate fraction of the skin washes yielded results in line with those obtained from the RCA staining of tissues. The four anuran species, reported to be *B. salamandrivorans* resistant or tolerant, showed a low percentage of free galactose in the total carbohydrate fraction (Fig. 6, Supplementary Fig. 2 and Supplementary Table 9). Moderate correlations were observed between the percentage of free galactose with *B. salamandrivorans* infection peak loads (Pearson $r = 0.641$, $p = 0.003$; Fig. 6a) and with mortality rates (Pearson $r = 0.523$, $p = 0.026$; Fig. 6b), though the sensitivity of free galactose likely varies across species, as suggested by *Lyciasalamandra helverseni*, *Salamandra salamandra* and *Calotriton asper* (Fig. 6a, b), which seem better able to tolerate higher free galactose levels than expected based on the observed linear correlations (i.e., observed infection peak loads and mortality rates outside the 95% CI for both Pearson's correlations). Quadratic regression models show that 70.4% of the variance in infection peak loads ($R^2 = 0.704$; Fig. 6a) and 54.3% of the variance in mortality rates ($R^2 = 0.543$; Fig. 6b) can be explained by the percentage of free galactose from the mucosome washes. Meanwhile, a clear correlation was also found between the galactose levels in mucosome washes and the RCA staining intensity, when comparing samples within the same species ($r_{pb} = 0.565$, $p = 0.035$; Fig. 6c). However, RCA scores account for a larger variation in the infection peak loads of *B. salamandrivorans* ($R^2 = 0.872$) than the method of measuring galactose in the mucosome washes ($R^2 = 0.704$). Therefore, although the galactose concentrations in skin washes can predict the infection intensity of *B. salamandrivorans* infection, we do not recommend using skin washes for predictions at species level if more invasive sampling is allowed.

**Intraspecies variation in carbohydrate pattern could promote selection towards increased resistance.** RCA scores were shown to vary significantly between individuals of some, but not all species examined. The coefficient of variation (CV%) for RCA scores of species *P. waltl* ($n = 11$), *I. alpestris* ($n = 12$) and *S. salamandra* ($n = 10$) were 18.56, 13.74 and 0.00%, respectively. Mortality rates after experimental exposure for these species are 61.5[6,19], 75[19,39] and 100%[6,19,39,41]. This variation in glycosylation patterns within species may be exploited to produce lineages with increased resistance against infection and disease, provided such characteristics are hereditary and do not incur harmful side effects such as decreasing the defensive capacity of the skin microbiome. In naturally infected

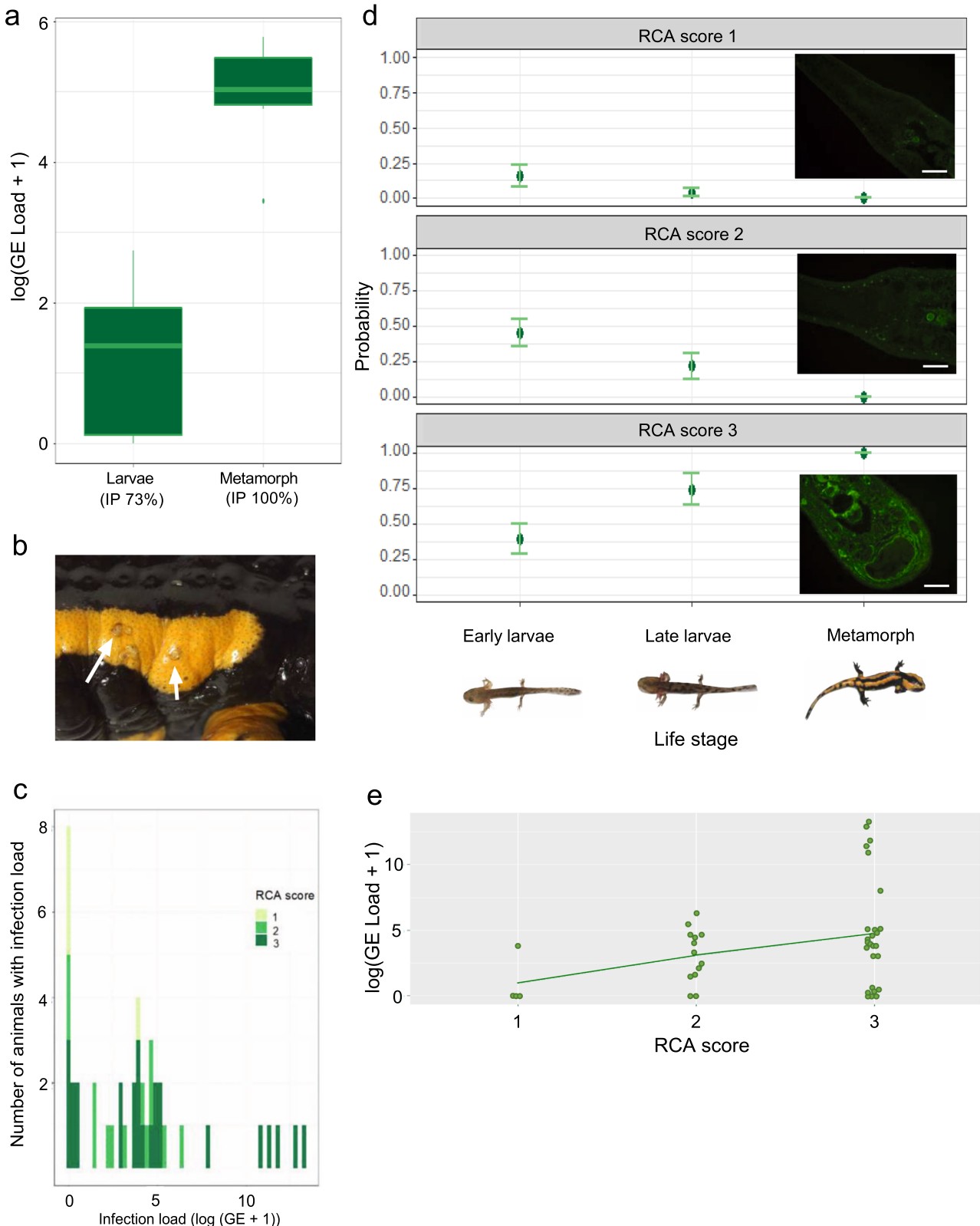

populations, such variation may result in the selection of increased resistance and thus may predict resilience against disease. Lack of variation in the galactose pattern observed in fire salamanders coincides with sharp population declines and the apparent lack of developing increased resistance observed in this species[39].

In this study, we were able to demonstrate that the interaction between host galactose-containing oligosaccharides and *B. salamandrivorans* ricin B-like lectin plays a vital role in the early pathogenesis of *B. salamandrivorans*-induced chytridiomycosis by mediating chemotaxis, adhesion, and the initiation of a virulent fungal response. Host cutaneous galactose content

**Fig. 4 Susceptibility to _B. salamandrivorans_ infection correlates with life stage dependent skin galactose presence. a** _B. salamandrivorans_ infection loads (expressed as genomic equivalents (GE)) on fire salamander larvae (_n_ = 37 biologically independent animals) and metamorphs (_n_ = 6 biologically independent animals) 10 days after exposure to 1.5 × 10⁵ spores/ml for 24 h. In the boxplots, horizontal lines represent median and interquartile ranges, with the vertical line representing min/max. Dots represent outliers, whiskers indicate highest/lowest value within 1.5*IQR from hinge. _B. salamandrivorans_ infection prevalence (IP) of fire salamander larvae and metamorphs are shown. **b** Macroscopic picture of infected fire salamander metamorphs, arrow indicates skin ulcerations. **c** Histogram of _B. salamandrivorans_ infection log (GE load + 1) load and equivalent RCA scores of fire salamander larvae and metamorphs. **d** Probability of RCA scores in early larvae stage (_n_ = 21 biologically independent animals), late larvae stage (_n_ = 16 biologically independent animals), and fire salamander metamorphs (_n_ = 6 biologically independent animals) predicted from ordinal logistical regression fit with polr(), error bars indicate RCA score probability +/− one standard error. Photomicrographs represent different RCA staining scores. RCA score: 1 = weak staining, 2 = strong staining, 3 = intense staining. RCA score of 0 is not shown because no slides were scored 0. Scale bars = 100 μm. Morphological characters of fire salamander larvae and metamorphs are shown. **e** _B. salamandrivorans_ infection log (GE load + 1) load and respective RCA scores of individual fire salamander larvae and metamorphs. The line indicates linear regression, implemented with RCA score as an ordered categorical variable. Source data are provided as a Source Data file.

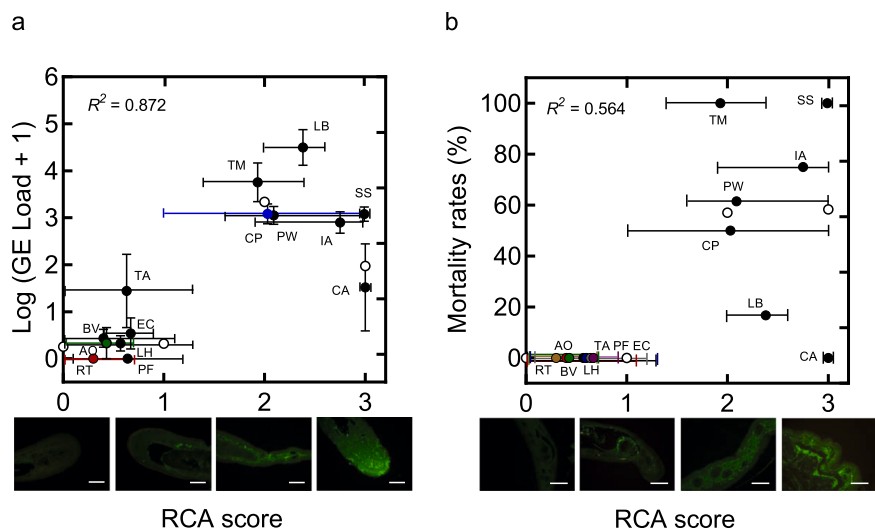

a

b

**Fig. 5 Skin galactose staining predicts _B. salamandrivorans_ infection intensity and survival probability. a** Intensity of RCA staining in relation to _B. salamandrivorans_ infection peak loads of different amphibian species. _B. salamandrivorans_ infection peak loads are expressed in log (genomic equivalents (GE) load +1). Vertical error bars indicate standard error of mean. **b** Intensity of RCA staining in relation to mortality rates of different amphibian species. **a**, **b** Horizontal error bars indicate minimum and maximum RCA scores, black and coloured dots represent the mean value overall. Linear regressions were performed using dummy variables for RCA scores. White dots represent the predicted values of infection peak loads and mortality rates for each RCA score generated by the regression model. _R²_ indicates the coefficient of multiple determination in the regression model. RCA score: 0 = negative staining, 1 = weak staining, 2 = strong staining, 3 = intense staining. Photomicrographs represent different RCA staining scores. Scale bars = 100 μm. Urodele species: LH = _Lissotriton helveticus_, PW = _Pleurodeles waltl_, LB = _Lissotriton boscai_, TA = _Triturus anatolicus_, TM = _Triturus marmoratus_, CP = _Cynops pyrrhogaster_, IA = _Ichthyosaura alpestris_, SS = _Salamandra Salamandra_ and CA = _Calotriton asper_. Anuran species: AO = _Alytes obstetricans_, RT = _Rana temporaria_, BV = _Bombina variegata_, PF = _Pelobates fuscus_ and EC = _Epidalea calamita_. Number of biologically independent animals used for RCA staining: LH (_n_ = 13), PW (_n_ = 11), LB (_n_ = 3), TA (_n_ = 3), TM (_n_ = 3), CP (_n_ = 3), IA (_n_ = 12), SS (_n_ = 10), CA (_n_ = 10), AO (_n_ = 10), RT (_n_ = 10), BV (_n_ = 5), PF (_n_ = 5) and EC (_n_ = 5). Number of biologically independent animals in infection trials: LH (_n_ = 23), PW (_n_ = 13), LB (_n_ = 6), TA (_n_ = 6), TM (_n_ = 6), CP (_n_ = 8), IA (_n_ = 20), SS (_n_ = 26), CA (_n_ = 5), AO (_n_ = 13), RT (_n_ = 5), BV (_n_ = 4), PF (_n_ = 5) and EC (_n_ = 5). Source data are provided as a Source Data file.

effectively predicted susceptibility to _B. salamandrivorans_ colonization. Intraspecific variation of galactose patterns may thus provide an opportunity for selection towards increased colonization resistance.

## Methods

**_Batrachochytrium salamandrivorans_ (_B. salamandrivorans_) culture conditions and zoospore isolation.** _B. salamandrivorans_ type strain (AMFP 13/01)⁸ was grown in tryptone-gelatin hydrolysate-lactose (TGhL) broth and incubated for 5−7 days at 15 °C. Zoospores were harvested by replacing the TGhL broth with distilled water. The collected water was filtered through a sterile mesh filter with pore size 10 μm (Pluristrainer, PluriSelect) to remove sporangia. Zoospore viability and mobility were confirmed using light microscopy.

**Salamander skin lysate binding assay.** Binding of _B. salamandrivorans_ spores to the protein or carbohydrate fractions from fire salamander (_Salamandra_

_salamandra_) skin was tested by treating fire salamander sloughed skin lysates enzymatically with glycoside hydrolases, followed by protein precipitation. An overview of the skin lysate binding assay is shown in Supplementary Fig. 3.

To collect the sloughed skin, ten captive-bred adult fire salamanders were housed at 15 ± 1 °C on moist tissue. The sloughed skin samples were ground with liquid nitrogen into a fine powder and then homogenized, using 3 ml RadioImmunoprecipitation assay (RIPA) buffer (Sigma-Aldrich) per gram of tissue. Samples were incubated for 1 h at 4 °C, centrifuged at 27.000 × _g_ for 10 min and the supernatant was subsequently collected. Protein concentration was determined using the Pierce™ BCA Protein Assay Kit (Thermo Fisher Scientific). The obtained skin lysate was equally divided, one part was treated with Protein Deglycosylation Mix II and two parts were kept as crude skin lysates. Protein Deglycosylation Mix II (New England BioLabs) was used to remove N-linked and O-linked glycans from glycoproteins. According to the manufacturer's instructions, 5 μl 10× Deglycosylation Mix Buffer I and 5 μl Protein Deglycosylation Mix II were added to 40 μl skin lysate. The mixture was incubated at 37 °C for 16 h. Protein precipitation was conducted on the redundant Protein Deglycosylation Mix II treated and crude skin lysates. The precipitation was performed by slowly adding

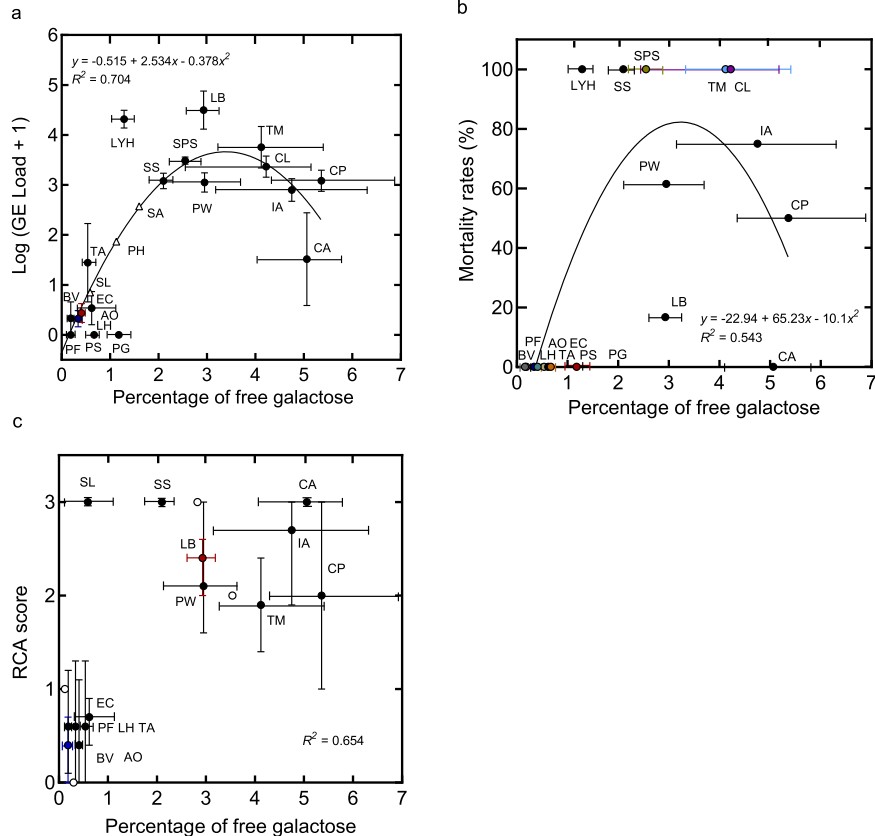

**Fig. 6 Free galactose levels as a biomarker for infection intensity. a** Percentage of free galactose per total carbohydrate in mucosome washes, calculated per square centimetre of body surface, in relation to *B. salamandrivorans* infection peak loads of different amphibian species. *B. salamandrivorans* infection peak loads are expressed in log (genomic equivalents (GE) load +1). Vertical error bars indicate standard error of mean. **b** Percentage of free galactose per total carbohydrate in mucosome washes, calculated per square centimetre of body surface, in relation to mortality rates of different amphibian species. **c** Percentage of free galactose per total carbohydrate in mucosome washes, calculated per square centimetre of body surface, in relation to intensity of RCA staining. Vertical error bars indicate minimum and maximum RCA scores. **a**, **c** Horizontal error bars indicate minimum and maximum of percentage of free galactose. Black and coloured dots represent the mean value overall. $R^2$ indicates the coefficient of multiple determination. **a**, **b** The regression analyses were performed by quadratic regression. White triangles represent the predicted infection peak load values calculated by the equation of quadratic regression model. Line indicates quadratic regression. **c** Linear regression was performed using dummy variables for RCA scores. White dots represent the predicted values of percentage of free galactose for each RCA score generated by the regression model. RCA score: 0 = negative staining, 1 = weak staining, 2 = strong staining, 3 = intense staining. **a**, **c** Urodele species: LH = *Lissotriton helveticus*, PW = *Pleurodeles waltl*, LB = *Lissotriton boscai*, TA = *Triturus anatolicus*, TM = *Triturus marmoratus*, CP = *Cynops pyrrhogaster*, IA = *Ichthyosaura alpestris*, SS = *Salamandra salamandra*, LYH = *Lyciasalamandera helverseni*, SPS = *Speleomantes strinatii*, PH = *Paramesotriton hongkongensis*, PG = *Plethodon glutinosus*, CL = *Chioglossa lusitanica*, PS = *Pachyhynobius shangchengensis*, CA = *Calotriton asper*, SA = *Salamandra algira* and SL = *Salamandra lanzai*. Anuran species: AO = *Alytes obstetricans*, BV = *Bombina variegata*, EC = *Epidalea calamita* and PF = *Pelobates fuscus*. Number of biologically independent animals used for measuring the percentage of free galactose in mucosome washes: LH ($n = 3$), PW ($n = 3$), LB ($n = 3$), TA ($n = 3$), TM ($n = 3$), CP ($n = 3$), IA ($n = 3$), SS ($n = 3$), LYH ($n = 3$), SPS ($n = 2$), PH ($n = 2$), PG ($n = 2$), CL ($n = 3$), PS ($n = 3$), CA ($n = 3$), SA ($n = 3$), SL ($n = 2$), AO ($n = 3$), BV ($n = 2$), EC ($n = 3$) and PF ($n = 3$). Number of biologically independent animals in infection trials: LH ($n = 23$), PW ($n = 13$), LB ($n = 6$), TA ($n = 6$), TM ($n = 6$), CP ($n = 8$), IA ($n = 20$), SS ($n = 26$), LYH ($n = 3$), SPS ($n = 3$), PG ($n = 5$), CL ($n = 6$), PS ($n = 3$), CA ($n = 5$), AO ($n = 13$), BV ($n = 4$), EC ($n = 5$) and PF ($n = 5$). Number of biologically independent animals used for RCA staining: LH ($n = 13$), PW ($n = 11$), LB ($n = 3$), TA ($n = 3$), TM ($n = 3$), CP ($n = 3$), IA ($n = 12$), SS ($n = 10$), CA ($n = 10$), AO ($n = 10$), RT ($n = 10$), BV ($n = 5$), PF ($n = 5$) and EC ($n = 5$). Source data are provided as a Source Data file.

saturated ammonium sulfate solution to the skin lysates to achieve a final concentration of 75%. Samples were then centrifuged at $21.130 \times g$ for 30 min to separate the precipitated proteins from the supernatant. The precipitated protein pellets were resuspended in 300 µl of 0.05 M carbonate−bicarbonate coating buffer (3.7 g NaHCO$_3$, 0.64 g Na$_2$CO$_3$, 1 L distilled water, pH 9.6). Each skin lysate solution was adjusted to the volume of 300 µl by adding a coating buffer. One hundred µl of each skin lysate solution was coated in each well of 96-well polystyrene microtiter plates (MaxiSorp$^{TM}$ plate, Thermo Fisher Scientific) in three technical replicates. As controls, coating buffer (negative control) and 75% ammonium sulfate solution were also coated on the 96-well plates. After incubation at 4 °C for 24 h the coated plates were washed three times with washing buffer (0.01 M PBS-Tween 20, pH 7.4) and blocked with 1% BSA overnight at 4 °C. Plates were then again washed three times with washing buffer and three times with distilled water. One hundred µl of *B. salamandrivorans* zoospore suspension ($1 \times 10^7$ zoospores per ml) were added in each well. Plates were incubated for

20 min at 15 °C and washed five times with distilled water to remove the unbound zoospores. Digital photographs were taken through via an inverted light microscope at 100 × magnification. Five pictures were taken for each well and zoospores in each photograph were counted in a blind fashion. Three independent repeats of the experiment were conducted (biological replicates).

**Carbohydrate binding assay.** To further determine which carbohydrates expressed on fire salamander sloughed skin can mediate the binding of *B. salamandrivorans* zoospores, *B. salamandrivorans* binding against four carbohydrates; N-acetylglucosamine (GlcNAc), N-acetylgalactosamine (GalNAc), mannose, and lactose was tested. The three monosaccharides and the disaccharide (Sigma-Aldrich) were dissolved and thereafter diluted in coating buffer to achieve a concentration of 5% (w/v). Then they were coated in triplicate wells by incubating at 4 °C for 24 h[42]. Plates were rinsed three times with washing buffer and blocked with 1% BSA overnight at 4 °C.

Hundred µl of *B. salamandrivorans* zoospore suspension ($1 \times 10^7$ zoospores per ml) was added in each well and incubated for 20 min at 15 °C. After washing the wells five times with distilled water to remove unbound zoospores, the plates were evaluated using a light microscope. Digital photographs were taken at 100 × magnification. Five pictures were taken for each well and zoospores in each photograph were counted in a blind fashion. Three independent repeats of the experiment were conducted (biological replicates).

In this experiment the highest level of *B. salamandrivorans* spores binding to lactose was observed. Lactose is a dissaccharide consisting of glucose and galactose. Therefore, in the following experiments galactose, glucose and their derivatives will be tested separately.

**Carbohydrate chemotaxis test**. Chemotaxis of *B. salamandrivorans* toward free carbohydrates was tested as previously explained (Supplementary Fig. 4)[12]. The sugars D-Glucose (Sigma-Aldrich), D-mannose (Sigma-Aldrich), Lactose (Sigma-Aldrich), and D-galactose (Sigma-Aldrich) were tested as attractant for *B. salamandrivorans*. The monosaccharides instead of the amide derivatives were used in this experiment to exclude any chemotactic signalling activity of the amides. Sugars were dissolved in distilled water, filter sterilized, and tested at a 0.1 M concentration. Hematocrit capillaries (75 mm length; Hirschmann laborgeräte, Eberstadt, Germany) were filled with 60 µl carbohydrate solution, vehicle control capillaries with 60 µl sterile distilled water. To prevent leakage, the capillaries were sealed with wax plugs (Hirschmann laborgeräte, Eberstadt, Germany) at one side. Each capillary was swiped on the outside with lens paper (Kimtech Science, Kimberley Clark, Roswell, GA, USA) to remove possible attractant spillover. Capillaries were incubated in 400 µl inoculum containing $10^6$ *B. salamandrivorans* zoospores in water and placed in a holder inclined about 65° upwards. The assay was incubated for 90 min at 15 °C, after which the capillaries were removed and swiped again at the outside to remove *B. salamandrivorans* zoospores possibly adhering on the outside. Inocula were checked for motility of the zoospores using an inverted microscope (Olympus CKX 41, Hamburg, Germany). Contents of the capillaries were collected and centrifuged for 2 min at $16.000 \times g$. The supernatant was removed as much as possible. The pellet was suspended in 100 µl Prepman Ultra Sample Preparation reagent (Applied Biosystems, Life Technologies Europe, Ghent, Belgium) and DNA was extracted according to the manufacturer's guidelines. For each sample, the number of *B. salamandrivorans* zoospores was quantified using quantitative real-time PCR (qPCR)[41], and data were analyzed using the Bio-Rad CFX manager 3.1. The primers and probe can be found in Supplementary Table 11. Within each assay, all carbohydrates and negative controls were tested at least in triplicate (technical replicates) and three independent repeats of the assay were performed (biological replicates).

**Carbohydrate transcriptome test**. RNA preparation: total RNA was isolated from *B. salamandrivorans* zoospores treated with different carbohydrates. Therefore, newly released zoospores (less than 2 h after induction of spore release by adding water) were harvested from 175 cm² cell culture flasks by replacing the TGhL broth with distilled water, which was filtered using a sterile mesh filter with pore size 10 µm (Pluristrainer, PluriSelect). Six-biological replicates containing $4 \times 10^7$ zoospores were obtained. Each biological replicate consisted of a pool of spores harvested from three cell culture flasks. Per biological replicate, the spores were divided into 4 eppendorfs ($10^7$ zoospores/eppendorf) which were treated for 1 h at 15 °C with $H_2O$ (control), 50 mM (D-galactose), 50 mM (D-glucose), or 50 mM (D-mannose) (Supplementary Fig. 5). After 1 h, the zoospores were centrifuged for 5 min at $4.000 \times g$ at 15 °C to remove the supernatant, after which RNA was extracted using the RNeasy mini kit (Qiagen)[18]. The RNA was treated with Turbo™ DNase (Ambion), the manufacturer's instructions. RNA degradation and contamination were monitored on 1% agarose gels. The RNA purity was checked using the NanoPhotometer® spectrophotometer (IMPLEN, CA, USA). Finally, the RNA integrity and quantitation were assessed using the RNA Nano 6000 assay kit of the Bioanalyzer 2100 system (Agilent Technologie, CA, USA).

Library preparation for transcriptome sequencing: Whole-transcriptome sequencing libraries were constructed and sequenced on the Illumina HiSeq platform (Novogen, China). A total amount of 1 µg RNA per sample was used as input material for the RNA sample preparations. Sequencing libraries were generated using NEBNext® Ultra™ RNA Library Prep Kit for Illumina® (NEB, USA) following the manufacturer's recommendations and index codes were added to attribute sequences to each sample. Briefly, mRNA was purified from total RNA using poly-T oligo-attached magnetic beads. Fragmentation was carried out using divalent cations under elevated temperature in NEBNext First Strand Synthesis Reaction Buffer (5X). First-strand cDNA was synthesized using random hexamer primer and M-MuLV Reverse Transcriptase (RNase H-). Second strand cDNA synthesis was subsequently performed using DNA Polymerase I and RNase H. Remaining overhangs were converted into blunt ends via exonuclease/polymerase activities. After adenylation of 3′ ends of DNA fragments, NEBNext Adaptor with hairpin loop structure was ligated to prepare for hybridization. In order to select cDNA fragments of preferentially 150−200 bp in length, the library fragments were purified with AMPure XP system (Beckman Coulter, Beverly, USA). Then 3 µl USER Enzyme (NEB, USA) was used with size-selected, adaptor-ligated cDNA at 37 °C for 15 min followed by 5 min at 95 °C before PCR. Then PCR was performed with Phusion High-Fidelity DNA polymerase, Universal PCR primers, and Index

(X) Primer. At last, PCR products were purified (AMPure XP system) and library quality was assessed on the Agilent Bioanalyzer 2100 system.

Clustering and sequencing: The clustering of the index-coded samples was performed on a cBot Cluster Generation System using PE Cluster Kit cBot-HS (Illumina) according to the manufacturer's instructions. After cluster generation, the library preparations were sequenced on an Illumina platform and paired-end reads were generated.

Quality analysis, mapping, and assembly: Raw data (raw reads) of FASTQ format were first processed through fastp (version 0.20.0). In this step, clean data (clean reads) were obtained by removing reads containing adapter and poly-N sequences and reads with low quality from raw data. At the same time, Q20, Q30, and GC content of the clean data were calculated (Supplementary Table 12). All the downstream analyses were based on the clean data with high quality. Reference genome and gene model annotation files were downloaded from genome website browser (NCBI/UCSC/Ensembl) directly. Paired-end clean reads were mapped to the *B. salamandrivorans* reference genome using HISAT2 (version 2.0.5) software[18]. Featurecounts (version 1.5.0-p3) were used to count the read numbers mapped to each gene, including known and novel genes (Supplementary Table 13). And then RPKM (reads per kilobase per million) of each gene was calculated based on the length of the gene and reads count mapped to this gene.

Gene expression, differential expression, enrichment, and coexpression- analysis: Differential expression analysis was performed using the DESeq2 R package[43]. The resulting $P$-values were adjusted using the Benjamini and Hochberg's approach for controlling the false discovery rate (FDR). Genes with an adjusted $P$-value < 0.05 found by DESeq2 were assigned as differentially expressed. Protein domains were annotated with PFAM version 27 and 33 and KEGG domains, Gene Ontology (GO) enrichment analysis of differentially expressed genes was implemented by the clusterProfiler R package[44] and dcGOR R package[45]. GO terms with corrected $P$-value less than 0.05 were considered significantly enriched by differential expressed genes. ClusterProfiler R package[44] was also used to test the statistical enrichment of differentially expressed genes in KEGG pathways.

**Detection of protease activity**. The influence of carbohydrate exposure on protease activity of *B. salamandrivorans* zoospores was assessed. Therefore, zoospores were harvested from 175 cm² cell culture flasks by replacing the TGhL broth with distilled water, which was filtered using a sterile mesh filter with pore size 10 µm (Pluristrainer, PluriSelect). A pool containing approximately $5 \times 10^7$ zoospores/ ml was obtained. 200 µl of the spore suspension ($10^7$ spores) was added to eppendorfs containing 200 µl $H_2O$ ($H_2O$; $n = 3$), 200 µl 100 mM D-Glucose (Glc; $n = 3$), 200 µl 100 mM D-mannose (Man; $n = 3$), 200 µl 100 mM D-galactose (Gal; $n = 3$), or as a control, 200 µl $H_2O$ containing protease inhibitor mix (P8215, Sigma-Aldrich) (PI; $n = 3$). After 1.5 h at 15 °C, the zoospores were centrifuged for 5 min at $4.000 \times g$ at 15 °C and the supernatant was collected. Protease activity in the supernatant was analyzed using the Pierce Fluorescent Protease Assay Kit (Thermo Fisher Scientific), according to the manufacturer's instructions. Three independent repeats of the experiment were performed (biological replicates).

**Identification of *B. salamandrivorans* lectin genes**. Potential candidates of carbohydrate-binding molecules (CBMs) were identified in the *B. salamandrivorans* (AMFP) genome listed in the NCBI database (Bioproject PRJNA311566).

*B. salamandrivorans* (AMFP 13/01) coding regions from the single annotated genome present on NCBI database (Bioproject PRJNA311566) were used to single out potential lectin genes of interest that could serve as genes of carbohydrate-binding proteins. The lectin candidates were identified with BLASTp (BLAST + 2.9.0) over the FungiDB database (constituting 199 candidates, database accessed 1st March 2018) using the stringent e-value cutoff of 1e−50 to avoid spurious hits[46,47].

From these, five candidates that referred to lectins and carbohydrate-binding were manually selected using the NCBI CDD (v3.16) conserved domain software with default settings[48].

Expression of two of these genes (BSLG_00833 and BSLG_02674) was confirmed by a previous mRNA expression analysis (Bioproject PRJNA311566)[18].

**Animals**. The animal experiments were performed following the European law and with the approval of the ethical committee of the Faculty of Veterinary Medicine (Ghent University EC) (EC2015/86). Only captive bred animals were used. Fire salamander larvae belonging to different life stages[49] were used in a *B. salamandrivorans* infection trial.

For lectin-histochemical staining, skin samples were collected from amphibian species *Salamandra salamandra* ($n = 10$), *Ichthyosaura alpestris* ($n = 12$), *Lissotriton helveticus* ($n = 13$), *Pleurodeles waltl* ($n = 11$), *Lissotriton boscai* ($n = 3$), *Alytes obstetricans* ($n = 10$), *Cynops pyrrhogaster* ($n = 3$), *Triturus anatolicus* ($n = 3$), *Triturus marmoratus* ($n = 3$), *Calotriton asper* ($n = 10$), *Bombina variegata* ($n = 5$), *Rana temporaria* ($n = 10$), *Epidalea calamita* ($n = 5$), *Pelobates fuscus* ($n = 5$) and *Salamandra lanzai* ($n = 3$). Tail or toe clips, ventral and dorsal skin samples were collected from animals that were euthanized with natrium pentobarbital 20% (KELA). The collected samples were immediately fixed in Bouin's solution for 24 h.

Mucosome samples were collected by bathing animals in HPLC-grade water for 1 h from 21 amphibian species (different animals as the ones used for the

tissueclips), namely *Lissotriton helveticus* (*n* = 3), *Pleurodeles waltl* (*n* = 3), *Lissotriton boscai* (*n* = 3), *Triturus anatolicus* (*n* = 3), *Triturus marmoratus* (*n* = 3), *Cynops pyrrhogaster* (*n* = 3), *Ichthyosaura alpestris* (*n* = 3), *Salamandra salamandra* (*n* = 3), *Lyciasalamandera helverseni* (*n* = 3), *Speleomantes strinatii* (*n* = 2), *Paramesotriton hongkongensis* (*n* = 2), *Plethodon glutinosus* (*n* = 2), *Chioglossa lusitanica* (*n* = 3), *Pachyhynobius shangchengensis* (*n* = 3), *Calotriton asper* (*n* = 3), *Salamandra algira* (*n* = 3), *Salamandra lanzai* (*n* = 2), *Alytes obstetricans* (*n* = 3), *Bombina variegata* (*n* = 2), *Epidalea calamita* (*n* = 3) *and Pelobates fuscus* (*n* = 3).

**Exposure of fire salamander larvae and metamorphs to *B. salamandrivorans*.** Twenty-two early-stage and 26 late-stage larvae[49,50] were inoculated with $1.5 \times 10^5$ *B. salamandrivorans* spores per ml water during 24 h. Ten days after the inoculation all the early-stage and sixteen late-stage larvae were euthanized. The two hind legs were analyzed by qPCR to detect the *B. salamandrivorans* GE load. A tail clip was stained with fluorescein-labelled RCA I (see below). Ten late-stage larvae were further kept until five weeks after metamorphosis.

Six one-week-old fire salamander metamorphs were inoculated with 1 ml of water containing $1.5 \times 10^5$ spores for 24 h. The animals were euthanized 10 days after inoculation. The two hind legs were analyzed by qPCR to detect the *B. salamandrivorans* GE load. A tail clip was stained with fluorescein labelled RCA I (see below).

**Lectin-histochemical staining.** Fluorescein labelled RCA I (*Ricinus communis* agglutinin I) (Vector Laboratories) and Con A (Concanavalin A) has been used to detect the expression of galactose and mannose or glucose in the epidermis of amphibians[38].

After 24 h fixation in Bouin's medium (Sigma-Aldrich), samples were washed first with tap water until the water ran colourless, then washed for 24 h in 70% ethanol saturated with lithium carbonate (Sigma-Aldrich) to remove picric acid. Tissues were then dehydrated in a graded ethanol series, cleared in xylene, embedded in paraffin, and sectioned in 4−6 μm slices. Before lectin staining, the sections were deparaffinized in xylene and hydrated in a series of ethyl alcohols. For better presenting the carbohydrate antigens, we performed antigen retrieval by submerging slides in citrate buffer (10 mM citric acid, pH 6.0) and heat treating in microwave (850 W for 3.5 min plus 450 W for 10 min). The slides were rinsed with PBS (0.01 M, pH 7.4) and immersed in 1% BSA (Sigma-Aldrich) for 15 min, to prevent non-specific lectin binding. Subsequently, the sections were incubated with either lectin RCA I (15 μg/ml) or lectin Con A (5 μg/ml) for 30 min. Lectins were diluted with lectin binding buffer (10 mM Hepes, 0.15 M NaCl, pH 7.5). As a negative control, lectin RCA I was mixed with 200 mM galactose, and lectin Con A was mixed with 200 mM mannose + 200 mM glucose, before incubating with skin sections to inhibit lectin binding. For positive control, a slide of fire salamander ventral skin sample for RCA staining, and midwife toad ventral skin sample for Con A staining, was included in each experiment. The slides were then washed in PBS, and cell nuclei were stained with 10 μg/ml Hoechst 33342 Solution (Invitrogen). Coverslips were mounted with Prolong™ Gold Antifade Reagent (Invitrogen). Staining results were observed using a Leica fluorescence microscope under 10× magnification, with a 450−490 nm BP excitation filter for lectin staining and a 355−425 nm BP excitation filter for Hoechst staining. Staining pictures were taken using Leica Application Suite (LAS) X software. The lectin staining intensities were classified as intense (3), strong (2), weak (1), or negative (0) staining (Supplementary Fig. 6). Experimental positive and negative controls were defined as intense (3) and negative (0) stained, respectively, and other slides were then evaluated in comparison to the set parameters. Hoechst staining results were paired with corresponding lectin staining results, making it easier to discern the tissue structure from the dark background. The fluorescent intensities were scored by three reviewers, respectively scoring the same dataset of pictures blinded three separate times, and the mean value was taken as the final result.

**Free galactose, mannose, and total carbohydrates in amphibian mucosome.** Mucus was collected from 21 amphibian species (see above). The animal body surface and volume of bathing water were calculated as follows: surface area of anuran species in cm² = 9.9* (mass in g)^(0.56), surface area of urodelan species in cm² = 8.42* (mass in g)^(0.694), and the quantity of HPLC-grade water to add to both anuran and urodelan species was determined by dividing the surface area by 4), and animals were bathed in respective amounts of HPLC-grade water for 1 h[40,51]. Animal washes were collected and concentrated by SpeedVac Vacuum Concentrators (Thermo Fisher Scientific) to 100 μl. The quantities of free galactose, mannose, and total carbohydrates in 100 μl of concentrated animal wash were measured using the Galactose Assay Kit (Abcam), Mannose ELISA Kit (Aviva Systems Biology), and Total Carbohydrates Assay Kit (Abcam), as per instructions. Concentrations of free galactose, mannose, and total carbohydrates in animal washes were divided by animal body surface to get the final results of sugar concentrations per square centimetre of the body surface.

**Statistical analysis.** Statistical analyses of fire salamander skin lysate binding assay, carbohydrate-binding assay, chemotaxis assay, and protease activity assay were

performed using R version 4.0.3. To account for the experimental design, Generalized Linear Mixed Models (GLMM, R library lme4[52]) were used, specifying a nested random effect whereby technical replicates are nested within biological replicates. Count data were modelled first using a Poisson distribution, but as significant over-dispersion was present in the data, a negative binomial error structure was implemented. For the protease activity assay, data do not represent counts and a log transformation on the raw values were used to ensure normality of model residuals (Shapiro-Wilk W > 0.95) allowing a Gaussian error structure (i.e., a Linear Mixed Model (LMM). To test for differences between categories, the (G)LMMs were directly fed to the glht function of the R library multcomp[53], setting up contrasts for Tukey's all-pair comparisons, resulting in Bonferroni-corrected *p*-values adjusted for multiple testing. Statistical analyses of the larvae infection trial were performed in R version 4.0.0, with tidyverse[54] version 1.3.0, MASS[55] version 7.3-51.6, VGAM[56] version 1.1-3, DHARMa[57] version 0.3.1 and glmmTMB[58] version 1.0.2.1. Infection loads of larvae and metamorphs were compared, using the Wilcoxon rank sum test, formula Chytrid GE load ~ larvae vs metamorph status, from the stats package. The correlation between larvae *Ricinus communis* agglutinin (RCA) scoring (1 = weak staining, 2 = strong staining, 3 = intense staining) and infection load was performed using the glm() function on log-transformed genomic equivalents with formula $\log_{10}$(*B. salamandrivorans* load in Genomic equivalents)~ RCA score, treating RCA score as an ordered factor with guassian distribution. As non-transformed chytrid loads showed zero-inflation and overdispersion, we also fit a generalized linear model with negative binomial distribution (GE load ~ RCA score) with RCA score as an ordered factor, using glmmTMB with a zero-inflation model (~ RCA score), which showed a comparable positive correlation between RCA score and GE load (conditional model coefficient = 5.67, *p* = 0.003, zero-inflation model coefficient = −2.21, *p* = 0.016). Residuals and chi-square test indicated the negative binomial model was not a significant improvement and so the simpler generalized linear model on transformed data was included. RCA scoring and larval stage prediction probabilities in Fig. 4c were generated by polr(RCA score ~ life stage) from MASS. Model fit and appropriateness was tested using Chisq test (*p* = 0.003), the model fit compared favourably to a more complex multinomial logit model and a model fit based on 70% of the data predicted 70−75% of remaining data (when data repeatedly sampled with different seeds, with the final model fit to all data).

The regression and correlation analyses of different amphibian species were performed in SPSS (IBM SPSS Statistics for Windows, Version 26.0. Armonk, NY, USA). Correlations of RCA scores with *B. salamandrivorans* infection peak loads, mortality rates, and percentage of free galactose were calculated by two-tailed Point-Biserial Correlation (*p* < 0.05 considered as significant), and the regression analyses were assessed by linear regression model with dummy variables. Dummy variables were generated from RCA scores, rounded to the nearest whole number. Correlations of percentages of free galactose with *B. salamandrivorans* infection peak loads and mortality rates were calculated using a two-tailed Pearson Correlation Coefficient test (correlation significant at *p* < 0.05), and regression analyses were assessed by a quadratic regression model.

**Reporting summary.** Further information on research design is available in the Nature Research Reporting Summary linked to this article.

## Data availability
All data reported in this study are provided in the Supplementary Information file and Supplementary Data 1. Source data are also provided with this paper. Carbohydrate binding genes were searched in the NCBI database, under Bioproject PRJNA311566. Reference genome and gene model annotation files used for RNA-seq annotation were downloaded from genome website browser NCBI (https://www.ncbi.nlm.nih.gov/), UCSC (http://genome.ucsc.edu/) and Ensembl (https://www.ensembl.org/index.html). RNA-seq data are available on the GEO website (http://www.ncbi.nlm.nih.gov/geo/) with Accession number GSE161129. Source data are provided with this paper.

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

## Acknowledgements

The technical assistance of Sarah Van Praet, Delphine Ameye, Joachim Christiaens, and Christian Puttevils is appreciated. This work was supported by the Ghent University Special Research Fund [BOF16/GOA/024] and Research Foundation Flanders [FWO Grants 12E6616N, 1507119N and G020318N].

## Author contributions

A.M., F.P., Y.W., and E.V. designed the study; Y.W., E.V., and A.M. conducted the experiments; M.K. delivered genetic data; Y.W., E.V., M.K., L.M., and D.S. performed the statistical analysis and figure generation; K.C. and N.C. contributed key study material; Y.W., E.V., A.M., and F.P. prepared the original paper. All authors reviewed the paper.

## Competing interests

The authors declare no competing interests.

## Additional information

**Peer review information** *Nature Communications* thanks Louise Rollins-Smith, Xavier Harrison and the other anonymous reviewer(s) of this work. Peer reviewer reports are available.

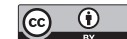

