## [Peer Review File · Nature Communications]

Reviewers' Comments:

Reviewer #1:

Remarks to the Author:

This is an interesting and salient manuscript that explores (1) the role of epidermal mucosal carbohydrates in *B. salamandrivorans* (Bsal) binding to amphibian hosts and (2) the correlation between the presence of carbohydrates and amphibian susceptibility to Bsal. The manuscript is well-written, enjoyable to read, and uses *in vitro* and *in vivo* approaches to investigate the interaction between Bsal and amphibian epidermal carbohydrates. Appropriate controls are included in the experimental designs and the experiments conducted progressed in a logical manner. I particularly liked the investigation of differential expression of Bsal genes exposed to different carbohydrates (RNAseq) as an added means to understand how Bsal initially responds to carbohydrates they may encounter in host epidermal mucus. Although this is a very good paper, I have a few major concerns with the study, primarily surrounding the statistical treatment of the data (which may or may not impact interpretation of the data) and the occasional over extension of correlative data to infer causation in specific wording of some sentences. Major concerns and minor editorial suggestions are detailed below.

Major Concerns:

1. The authors conflate pseudoreplicates (technical replicates) and biological replicates and are treating all measurements as independent values in their statistical analysis. This is incorrect and statistical analysis of the data need to be reperformed. Technical replicates estimate the variation or error within the technique and thus are likely to be very close to each other in value (i.e. not independent of each other). Conversely, biological replicates provide us with an idea of the biological variation that exists within the population and can be considered (for the most part, depending on how experiment being conducted) largely independent of each other. For example, in the skin lysate binding assay, the authors have 300 ul of solution obtained from one sample that they pipette into three individual wells (100 ul each), and then they conducted the experiment three independent times. The authors are stating that their number of independent tests is $n = 9$, when really it is $n = 3$. Treatment of data in this manner will have a large effect on the statistical analyses of the data and potentially on the interpretations that the authors make in regards using the outputs of these statistical analyses. In addition, all figures should be updated to simply represent the mean and variation of the biological replicates. These changes should be incorporated in the following results/discussions sections, figures, supplementary materials, and figure legends:

1a. Results described in the section on "Galactose mediates Bsal attraction and adhesion to salamander skin" (lines 75 – 113), Figure 1A, B, C, Supplementary Table 2, Supplementary Table 3, and Supplementary Table 5. As outlined in the above paragraph, the authors are treating this data as if they have $n = 9$, when they have $n = 3$ biological replicates.

1b. Results described in the section on "Contact with galactose spurs fungal virulence" and corresponding figures - it is unclear whether the authors harvested a large batch of zoospores at one point in time from multiple cultures of Bsal and from this made separate "pools" of zoospores which were further subdivided across treatments OR if the authors cultured zoospores on multiple different days and used the zoospores collected on those different days to perform independent experiments (methods described on Line 372 – 380). If multiple flasks were cultured and harvested all on one day (all seeded from the same culture), then I would consider those "pools" of 4×10^7 zoospores to be technical replicates and not biological replicates. However, culturing a set of flasks of zoospores for use in an independent trial on day 0, then another set of flasks for use on day 3, on day 6, etc. could be considered independent replicates.

1c. The results section on "Detection of protease activity" and corresponding Figure 3: Similar to above, it appears as though both technical and biological replicates are included in the determination of the number of independent replicates (i.e. $n = 9$) instead of the number of true independent replicates (i.e. $n = 3$ biological replicates). The provision of data for these experiments in a supplementary table is also lacking.

1d. Fig. 4 – Fig. 6 and accompanying results: It is unclear in the methods and data provided for

these figures what is considered technical versus biological replicates. In consideration of the concerns regarding treatment of data in statistical analysis, the authors should examine their data carefully to identify what data represent true biological/independent replicates for use in statistical analysis and reperform statistical analysis as needed. It should be clearly noted what is considered a biological replicate in these experiments and for the purposes of statistical testing.

2. Is there a need to normalize the data to a reference group within each trial for the experiments determining how many Bsal spores bind to skin lysates or commercially purchased carbohydrates? The reason for this inquiry/suggestion is two fold: (1) By normalizing a group within a single trial to 100%, the biological/independent variation of this group across the independent trials is lost. Can the authors comment on how they adjusted for this lack of variation, and presumably unequal variation across the treatment groups, during statistical analysis of the data? (2) Normalization of the data also masks the total number of spores that bound to the well. For example, the authors added one million zoospores to each well, but only a small percent of spores (less than 0.3%) bind to the crude skin lysate. I find it interesting that such a small proportion of the zoospores bind to the different carbohydrate treatment groups. Is this reflective of the proportion of zoospores that bind carbohydrates in the skin or reflective of insufficient carbohydrate binding sites in the coated well? Furthermore, Bsal spores do not bind to individual carbohydrates (1130 – 1917, Supplementary Table 5) to the same extent as crude skin lysates (mean = 2878, Supplementary Table 2). This nuance also gets lost with normalization.

3. The authors analyzed the data in Figure 3 using a Welch t-test followed by a Tukey's multiple comparison post hoc test. This statement seems problematic for two reasons: (1) more than two groups are being compared, which suggests the data should be analyzed using a multiple comparison test instead of using multiple t-tests (increases the chance of type I error), and (2) post hoc tests do not need to be performed after t-tests.

4. The authors performed transcriptome analysis on Bsal spores exposed to water, galactose, glucose or mannose, identifying 25 genes that were differentially expressed in Bsal spores exposed to galactose. The authors nicely elaborated on these differentially regulated genes and their functions. Similarly, 26 genes were uniquely differentially regulated in response to mannose and 52 were uniquely differentially regulated in response to glucose, however these differentially regulated genes were not discussed. Providing a description of the nature and role of these differentially regulated genes in response to other carbohydrates may provide better support for the unique induction of virulence genes in Bsal exposed to galactose.

5. The authors provide convincing evidence of a correlation between presence/abundance of galactose in the skin mucosome and Bsal colonization intensity. While the authors are careful to state their findings are a correlation, it is important that causation is not inferred from correlation and I suggest adding in a brief paragraph or a few sentences to the discussion or conclusion that addresses this point and explore alternative possibilities. I think this is an important addition to the manuscript as on Lines 211 – 214 the authors state "we found that galactose content in the skin is a poor predictor of disease course after infection. Indeed, extensive colonization needs not necessarily result in lethality. Rather, host defense mechanisms will determine the disease outcome." I have suggested a few places in the manuscript (outlined in the minor comments) that wording can be changed to ensure that colonization of a host does not imply susceptibility in recognition that host susceptibility is a complex process mediated by host, pathogen and environment factors.

Minor editorial suggestions:

Line 21: Please clarify that mitigation measures for chytridiomycosis have been "largely unsuccessful" versus work done to understand the mechanisms underpinning disease ecology.

Lines 23 -24: "the first known marker of host susceptibility" – should be edited to "the first known marker of host colonization". This suggestion is based on the later discussion in the manuscript where authors indicate that presence of galactose is not an accurate predictor of "disease course after infection". Accordingly, all other instances in the manuscript should be adjusted to reflect the difference between colonization and susceptibility of a host.

Lines 25 – 28, 70 - 72: Suggest rewording from “susceptibility” to “colonization” to better represent the findings presented in the manuscript.

Lines 39 – 40: The authors cite Scheele et al. A conflicting view of this data was raised in the Comment paper by Lambert et al 2000 (Science, 367: 1838) regarding the number of species identified in the p

The authors should consider adding a brief paragraph to the introduction to describe the carbohydrate structures that have been identified in frog epidermal mucous and the role of these carbohydrate in host-pathogen interactions. The reader’s understanding of the experiments that follow are reliant on their understanding of this topic.

Lines 40 – 42: Recent studies have also explored the use of commensal skin bacteria that produce antifungal metabolites as probiotics and may be worth mentioning here.

Line 49: Consider identifying the salamander species (fire salamanders) involved in the collapsing population in Netherlands – this would provide some context for the species being used in this study.

The authors observed the highest level of Bsal spores binding to lactose (Line 94 and Fig. 1B), however, lactose was not examined in the following experiments. Perhaps the authors can provide a statement indicating the logic behind this experimental choice?

Lines 92 – 99: The authors use GlcNAc, lactose, mannose and GalNAC in the binding experiment but then use galactose, lactose, mannose and glucose in the chemotaxis experiment. Please add a statement explaining the choice to switch from GlcNAc to glucose and from GalNAC to galactose.

Centrifugal forces should be reported in x g instead of rpm. (e.g. Line 312). Please correct here and elsewhere.

When reporting magnification, the authors should also account for the magnification of the ocular lens (e.g. Line 326). Please correct here and elsewhere.

Lines 505 – 507: A reference for previous studies demonstrating the ability of RCA I and Con A to detect carbohydrates of amphibians should be provided.

Supplementary Figure 1: Due to the lack of fluorescent signal in the images, it is difficult to verify the presence of skin tissues. Perhaps the authors could include some accompanying light images of the sectioned skin tissues?

Supplementary Figure 4. Consider revising the title of the figure so that it is apparent to which experiment these values correspond.

Supplementary Table 9: The table legend should include details about what “-” indicates (not tested? not detected?) as this symbol appears in multiple instances in the table.

Abstract and Conclusion Sections: The authors suggest that selective breeding to reduce the presence of galactose carbohydrates in the mucosome is a viable strategy to increase disease resistance and thus conserve susceptible amphibian species. However, the authors should discuss the potential limitations of this approach as well – to my knowledge, the contribution of glycosylation modifications to the function of amphibian epidermal mucus in pathogen defense and the role these carbohydrates play in selection of commensal bacteria (that may produce antifungal metabolites) are largely unexplored. In addition, the authors should be cautious in making this statement given that they acknowledge that presence of galactose is not an accurate predictor of “disease course after infection”

Reviewer #2:

Remarks to the Author:

Overview:

Noteworthy Results: This is a well-written, technically sound, and original study that provides new insights into the mechanism by which the chytrid pathogen, *Batrachochytrium salamandrivorans*, infects new hosts. By identifying galactose as a key sugar displayed on skin proteins that appears to attract swimming zoospores and allows them to attach, the results suggest an explanation for why some species of amphibians are more susceptible to infection and disease and why salamander larvae are less susceptible than adults of the same species (low expression of galactose in the skin). RNAseq analysis of the expression profile of zoospores exposed to galactose showed the differential upregulation of a number of genes in comparison with zoospores in water alone. Some of the upregulated genes encode putative virulence factors such as M36 metalloproteases.

Significance to the Field of Amphibian Conservation: Levels of galactose expression in the skin washes (mucosome) are correlated with level of *B. salamandrivorans* infection, but more importantly, this observation may suggest a non-invasive method to assess whether other species would be more or less susceptible. That is, a quick assay of the amount of galactose in the mucosome might provide useful information for conservationists and wildlife managers to determine which amphibian species would be most vulnerable to infection and disease caused by this pathogen.

Overall assessment: This is a sound study that will advance our understanding of the pathogenesis of *B. salamandrivorans*. My comments are mostly suggestions for minor changes to add additional detail to the methods.

Specific Suggestions:

Major:

1. Line 299. Please indicate that RIPA buffer is Radioimmunoprecipitation assay buffer. Was it made in house or a commercial preparation?
2. Line 314. Please define the coating buffer more precisely. Is it 3.7 g Sodium Bicarbonate (NaHCO₃) and 0.64 g Sodium Carbonate (Na₂CO₃) per 1 L of distilled water?
3. Lines 325-328. Please indicate whether counts of attached zoospores were done in a blinded fashion.
4. Lines 342-344. Please indicate whether the counts of attached zoospores were done in a blinded fashion.
5. Lines 346-370. I have read the description of this chemotaxis assay in the original paper (Van Rooij et al. 2015) and again here, and I still can't quite understand it. Is the open end of the capillary in the 400 µl of zoospore solution? Does the closed end stick up and out of the solution at an angle? Since this is a critical assay, it is important to understand. Thus maybe a small schematic diagram of this assay in the supplement would be very helpful for anyone who might like to repeat this assay.
6. Lines 378-380. Please comment somewhere in the methods or discussion. This is a very short zoospore treatment period. Did you look at any other time points? Were the zoospores newly released? How long after release from the zoosporangium before they settle and form a germ tube? How much of gene expression could be due to "encystment" preparation than invasion or expression of other virulence factors. How clean were the zoospores (no more advanced thalli present)?
7. Line 445. Please add the temperature of incubation. I assume it was 15°C.
8. Lines 482-490. Please briefly describe how the mucosome samples were collected and processed.
9. Lines 541-542. Please indicate the volume of water per area of skin for mucosome collection.

Minor:

1. Line 316. I suggest this minor change to the wording "One hundred µl aliquots...".
2. Line 322. Change to "One hundred".
3. Line 505. Please indicate the species name (*communis*) of *Ricinus communis* in lower case and italics.
4. Line 352. Add the word "with", i.e., "filled with 60 µl carbohydrate".
5. Line 422. Please define RPKM as Reads per kilobase of transcript per million mapped reads.

6. Line 561. Replace the word "normally" with "normal".
7. Line 567. Define the RCA scoring here again. Remind the readers at this point of the nature of the lectin, *Ricinus communis* agglutinin.

Reviewer #3:

Remarks to the Author:

Review of Wang et al Nat Comms

Here the authors investigate the interaction between chemical composition of the amphibian skin and susceptibility to a lethal fungal pathogen. They employ a range of approaches including i) assays of ii) multi-species experimental infections to quantify the link between levels of galactose in host skin and subsequent infection success and progression; and iii) RNAseq to quantify pathogen gene regulation changes in response to host contact.

This is a very interesting paper with some important findings. The methodological approach is well laid out and logical to follow. The authors clearly describe how they narrowed down the target pool of molecules from 'protein or carbohydrate?' to 'which carbohydrate?', as well as cross-checking the pathogen genome for the respective carbohydrate-binding genes.

My only concern about the manuscript is that, as presented, it's very difficult to assess how robust the modelling of the data are. Throughout it's clear that there are both biological and technical replicates, which the authors do explain clearly. But I worry that these technical replicates are being fed into models and treated as independent degrees of freedom.

For example, in your skin lysate binding assay (L291-), 10 newts have their sloughed skin collected. You perform 3 independent experiments with 3 technical replicates per experiment. But each of these 9 data points can only ever come from 1 newt's sample. So they are all technical replicates and should be treated as such. Related to this, if feeding these individual data points into models, t tests (Fig 3) and Kruskal-Wallis tests (Fig 1) become inappropriate.

In addition, lots of the modelling for multi-species experiments seems to pool data from multiple species in linear modelling frameworks, so not in fact accounting for the fact that residuals are not exchangeable between species. Collectively these approaches, if I've understood correctly, will overinflate perceived statistical power. Some clarification here will help because if I've misunderstood then others will too.

Line 565: missing version number for glmmTMB. Also missing citations for all of these packages

Rebuttal letter

We would like to thank the reviewers for their time and constructive comments which we think have led to a much-improved manuscript.

The line numbers mentioned in our rebuttal letter below refer to the new line numbers in the manuscript. All changes in the manuscript text file are shown with track changes.

Reviewer #1 :

Major Concerns:

1. The authors conflate pseudoreplicates (technical replicates) and biological replicates and are treating all measurements as independent values in their statistical analysis. This is incorrect and statistical analysis of the data need to be reperformed. Technical replicates estimate the variation or error within the technique and thus are likely to be very close to each other in value (i.e. not independent of each other). Conversely, biological replicates provide us with an idea of the biological variation that exists within the population and can be considered (for the most part, depending on how experiment being conducted) largely independent of each other. For example, in the skin lysate binding assay, the authors have 300 ul of solution obtained from one sample that they pipette into three individual wells (100 ul each), and then they conducted the experiment three independent times. The authors are stating that their number of independent tests is $n = 9$, when really it is $n = 3$. Treatment of data in this manner will have a large effect on the statistical analyses of the data and potentially on the interpretations that the authors make in regards using the outputs of these statistical analyses. In addition, all figures should be updated to simply represent the mean and variation of the biological replicates. These changes should be incorporated in the following results/discussions sections, figures, supplementary materials, and figure legends:
>> We have overhauled these analyses by adopting a Generalized Linear Mixed Model (GLMM) that explicitly accounts for the nested design of these analyses (i.e. technical replicates nested within biological replicates), and directly modelling the raw count data (i.e. no normalization of data). For details, please see our replies to the comments below, but overall this reanalysis did not lead to important changes to our results and interpretation. Figures have been updated to present the mean and variation of the biological replicates, as suggested by the reviewer. Material and methods is changed accordingly at lines 589-600.

1a. Results described in the section on “Galactose mediates Bsal attraction and adhesion to salamander skin” (lines 75 – 113), Figure 1A, B, C, Supplementary Table 2, Supplementary Table 3, and Supplementary Table 5. As outlined in the above paragraph, the authors are treating this data as if they have $n = 9$, when they have $n = 3$ biological replicates.

>> The analyses corresponding to Figure 1(a-c) have been redone using GLMMs with ‘well’ (technical replicate) nested within ‘experiment’ (biological replicate) as a nested random effect. Count data were first modelled using a Poisson count distribution but as considerable overdispersion was present in the data, a negative binomial error structured provided a better model fit and was subsequently used. To test for differences in the number of zoospores bound to the different microtiter plates treatments, contrasts were set up using the general linear hypothesis test ‘glht’ function of the R library ‘multcomp’ (multiple comparisons), using Tukey’s all-pair comparisons correction, resulting in Bonferroni-corrected p-values adjusted for multiple testing. Figure 1(a-c) is replaced by the new Figure1 (a-c), we corrected supplementary table 2 and added supplementary tables 4 and 5.

1b. Results described in the section on “Contact with galactose spurs fungal virulence” and

corresponding figures - it is unclear whether the authors harvested a large batch of zoospores at one point in time from multiple cultures of Bsal and from this made separate “pools” of zoospores which were further subdivided across treatments OR if the authors cultured zoospores on multiple different days and used the zoospores collected on those different days to perform independent experiments (methods described on Line 372 – 380). If multiple flasks were cultured and harvested all on one day (all seeded from the same culture), then I would consider those “pools” of 4×10^7 zoospores to be technical replicates and not biological replicates. However, culturing a set of flasks of zoospores for use in an independent trial on day 0, then another set of flasks for use on day 3, on day 6, etc. could be considered independent replicates.

>> Per independent trial we used another set of flasks (independent replicates). Because the high amount of spores needed, we used per independent trial three flasks. We clarified this in the text and rephrased the sentence ‘six-independent pools containing 4×10^7 zoospores were obtained (biological replicates)’. The new sentences on lines 404-407 are: ‘Six-biological replicates containing 4×10^7 zoospores were obtained. Each biological replicate consisted of a pool a spores harvested from three cell culture flasks. Per biological replicate, the spores were divided in 4 eppendorfs...’

1c. The results section on “Detection of protease activity” and corresponding Figure 3: Similar to above, it appears as though both technical and biological replicates are included in the determination of the number of independent replicates (i.e. $n = 9$) instead of the number of true independent replicates (i.e. $n = 3$ biological replicates). The provision of data for these experiments in a supplementary table is also lacking.

>> These analyses follow a similar format as above, whereby indeed three independent experiments were performed, and each of these biological replicates has three technical replicates. Analyses have been redone as above (corrected on Lines 170-172) (please see reply to, though using Linear Mixed Models with a Gaussian error distribution and log-transformed raw values. We added Supplementary Table 7. Raw data are presented in Supplementary Dataset 1.

1d. Fig. 4 – Fig. 6 and accompanying results: It is unclear in the methods and data provided for these figures what is considered technical versus biological replicates. In consideration of the concerns regarding treatment of data in statistical analysis, the authors should examine their data carefully to identify what data represent true biological/independent replicates for use in statistical analysis and reperform statistical analysis as needed. It should be clearly noted what is considered a biological replicate in these experiments and for the purposes of statistical testing.

>> These figures refer to different analyses than the ones presented above, and do not follow the biological/technical replicates format. Figure 4: each datapoint represent a different individual and hence a linear model is justified (i.e. no need for mixed models). Figure 5 and Figure 6: please see our reply to a comment of the second reviewer to these analyses (pertaining to the treatment of data pertaining to multiple species).

2. Is there a need to normalize the data to a reference group within each trial for the experiments determining how many Bsal spores bind to skin lysates or commercially purchased carbohydrates? The reason for this inquiry/suggestion is two fold: (1) By normalizing a group within a single trial to 100%, the biological/independent variation of this group across the independent trials is lost. Can the authors comment on how they adjusted for this lack of variation, and presumably unequal variation across the treatment groups, during statistical analysis of the data? (2) Normalization of the data also masks the total number of

spores that bound to the well. For example, the authors added one million zoospores to each well, but only a small percent of spores (less than 0.3%) bind to the crude skin lysate. I find it interesting that such a small proportion of the zoospores bind to the different carbohydrate treatment groups. Is this reflective of the proportion of zoospores that bind carbohydrates in the skin or reflective of insufficient carbohydrate binding sites in the coated well? Furthermore, Bsal spores do not bind to individual carbohydrates (1130 – 1917, Supplementary Table 5) to the same extent as crude skin lysates (mean = 2878, Supplementary Table 2). This nuance also gets lost with normalization.

>> Thank you for pointing this out. We now re-analyzed all data (see above) without normalization, but using the raw data (negative binomial distribution for count data, Gaussian for continuous data).

We included a standard condition as a reference treatment, a positive control for spore binding. While care is taken to standardize all experimental steps, binding assays using multiple biological compounds (eg skin lysate, fungal spores) tend to show assay dependent variation. Comparing to a standard condition supports comparing results across different assays (eg different well plates).

We agree with this reviewer that we give the impression that a minor fraction of the spores is bound and we apologize for this lack of clarity. We mislabeled the results ("number of spores bound per well"), which should actually read: "spores per microscopic view of field", representing the average number of spores counted in 5 microscopic fields of view (which was a horrendous undertaking...). This has been adapted in the figure and in the supplementary dataset 1.

3. The authors analyzed the data in Figure 3 using a Welch t-test followed by a Tukey's multiple comparison post hoc test. This statement seems problematic for two reasons: (1) more than two groups are being compared, which suggests the data should be analyzed using a multiple comparison test instead of using multiple t-tests (increases the chance of type I error), and (2) post hoc tests do not need to be performed after t-tests.

>> Thank you for pointing this out. We have now redone these analyses by directly importing the linear mixed models into the 'glht' function of R package 'multcomp', and using Tukey all comparisons correction for obtaining p-values adjusted for multiple testing.

4. The authors performed transcriptome analysis on Bsal spores exposed to water, galactose, glucose or mannose, identifying 25 genes that were differentially expressed in Bsal spores exposed to galactose. The authors nicely elaborated on these differentially regulated genes and their functions. Similarly, 26 genes were uniquely differentially regulated in response to mannose and 52 were uniquely differentially regulated in response to glucose, however these differentially regulated genes were not discussed. Providing a description of the nature and role of these differentially regulated genes in response to other carbohydrates may provide better support for the unique induction of virulence genes in Bsal exposed to galactose.

>> We have added a table which includes the annotations of these genes (Supplementary Dataset 1), and added discussion to Lines 139-147: "Gene groups uniquely expressed in mannose- and glucose-treated spores contained repair enzymes, proteins linked to transport, but also possible virulence candidates including protein kinase (mannose BSLG_00482 and glucose BSLG_08672) and copper/zinc superoxide dismutase (mannose BSLG_09793), which is known as a superoxide radical scavenger linked to fungal virulence³⁶. Genes uniquely expressed in mannose were mainly associated with mitochondria, whereas the majority of the genes uniquely expressed in glucose seemed to be associated with the transport of cellular components and particularly proteins (e.g. secretory exocyst component, kinesin motor domain, transporters, nexin, clathrin, exportin) (Supplementary Dataset 1)."

5. The authors provide convincing evidence of a correlation between presence/abundance of galactose in the skin mucosome and Bsal colonization intensity. While the authors are careful to state their findings are a correlation, it is important that causation is not inferred from correlation and I suggest adding in a brief paragraph or a few sentences to the discussion or conclusion that addresses this point and explore alternative possibilities. I think this is an important addition to the manuscript as on Lines 211 – 214 the authors state “we found that galactose content in the skin is a poor predictor of disease course after infection. Indeed, extensive colonization needs not necessarily result in lethality. Rather, host defense mechanisms will determine the disease outcome.” I have suggested a few places in the manuscript (outlined in the minor comments) that wording can be changed to ensure that colonization of a host does not imply susceptibility in recognition that host susceptibility is a complex process mediated by host, pathogen and environment factors.
>> Thank you for this suggestion. We added extra information on the possible mechanisms in the sentence on Line 228. The new sentence is: “Rather, innate and acquired defense mechanisms and the environmental context are likely to determine the disease outcome.”

Minor editorial suggestions:

Line 21: Please clarify that mitigation measures for chytridiomycosis have been “largely unsuccessful” versus work done to understand the mechanisms underpinning disease ecology.
>> We agree with the referee that this needs clarification, yet it would take up too much space in the abstract and we propose to omit “for chytridiomycosis such efforts have thus far proven largely unsuccessful” from the abstract.

Lines 23 -24: “the first known marker of host susceptibility” – should be edited to “the first known marker of host colonization”. This suggestion is based on the later discussion in the manuscript where authors indicate that presence of galactose is not an accurate predictor of “disease course after infection”. Accordingly, all other instances in the manuscript should be adjusted to reflect the difference between colonization and susceptibility of a host.

>> We edited susceptibility to colonization. Based on this suggestion we also edited:

Line 2, 26 and 29: susceptibility changed to colonization

Line 30: to fungal colonization added

Lines78: infection susceptibility edited to fungal colonization

Line 175 colonization is added

Line 296 disease is replaced by colonization

Lines 25 – 28, 70 - 72: Suggest rewording from “susceptibility” to “colonization” to better represent the findings presented in the manuscript.

>> Lines 26, 29, 78: susceptibility is reworded to colonization

Lines 39 – 40: The authors cite Scheele et al. A conflicting view of this data was raised in the Comment paper by Lambert et al 2000 (Science, 367: 1838) regarding the number of species identified in the p

>> We rephrased the estimated number of affected species more carefully. We modified the sentence to ‘This fungal disease is linked to extinctions or declines of hundreds amphibian species worldwide’ and also cited the comment paper of Lambert et al., 2020 (that in turn was rebutted by Scheele et al. but perhaps it is best to leave it to the reader to go through the whole discussion). We do agree that the exact number of species is open for debate, yet we are convinced that chytridiomycosis is a major driver (or co-driver) of global amphibian declines.

The authors should consider adding a brief paragraph to the introduction to describe the carbohydrate structures that have been identified in frog epidermal mucous and the role of these carbohydrate in host-pathogen interactions. The reader's understanding of the experiments that follow are reliant on their understanding of this topic.

>> Thank you for this comment. We briefly introduced the (known) role of carbohydrates in the chytrid – host interaction, as well as the term “mucosome” (Lines 65-68). Carbohydrate compounds of importance are being introduced Lines 105-108.

Lines 40 – 42: Recent studies have also explored the use of commensal skin bacteria that produce antifungal metabolites as probiotics and may be worth mentioning here.

>> ‘probiotics’ is added to the sentence on Line 43. Reference of McKenzie et al. 2020 who reviewed this topic is added to the reference list. For Bsal, the choice for probiotics seems less obvious, given the paucity of resident bacteria on healthy salamanders (in this case fire salamanders, *Salamandra salamandra*), in which beneficial effects of probiotic bacteria could be obtained only after adding high numbers, which rapidly decreased when probiotic treatment ended (Bletz et al., 2018, Proceedings of the Royal Society B – Biological Sciences, 285: article 20180758).

Line 49: Consider identifying the salamander species (fire salamanders) involved in the collapsing population in Netherlands – this would provide some context for the species being used in this study.

>> We added the salamander species to the sentence (Line 50-51)

The authors observed the highest level of Bsal spores binding to lactose (Line 94 and Fig. 1B), however, lactose was not examined in the following experiments. Perhaps the authors can provide a statement indicating the logic behind this experimental choice?

>> Lactose is a disaccharide, consisting of glucose and galactose. Therefore, it is not clear whether spores binding to lactose interact with galactose or glucose. In the following experiments we used galactose, glucose and their derivatives separately. For clarification, we added this information in the materials and methods section on Lines 367-369.

Lines 92 – 99: The authors use GlcNAc, lactose, mannose and GalNAC in the binding experiment but then use galactose, lactose, mannose and glucose in the chemotaxis experiment. Please add a statement explaining the choice to switch from GlcNAc to glucose and from GalNAC to galactose.

>> GlcNAc is an amide derivative of the glucose and GalNAC is an amide derivative of galactose. Mannose doesn't have an amide derivative. It has been reported that lectins bind more strongly to sugar derivatives than with a simple monosaccharide (Ramos et al., 2000), therefore, we chose GlcNAc and GalNAC for binding experiments.

Galactose-binding lectins also bind to GalNAC, and Glucose-binding lectins also bind to GlcNAc. However, we used monosaccharides in the chemotaxis experiment to exclude any chemotactic signalling of the amides. A statement explaining the choice to switch is added to the material and method section on Lines 375-377.

Centrifugal forces should be reported in x g instead of rpm. (e.g. Line 312). Please correct here and elsewhere.

>> rpm is changed to x g in Lines 331, 390, 410, 477.

When reporting magnification, the authors should also account for the magnification of the ocular lens (e.g. Line 326). Please correct here and elsewhere.

>> We corrected the magnification on Lines 346 and 363.

Lines 505 – 507: A reference for previous studies demonstrating the ability of RCA I and Con A to detect carbohydrates of amphibians should be provided.

>> The reference of Zaccone et al., 1999 is added to Line 540.

Supplementary Figure 1: Due to the lack of fluorescent signal in the images, it is difficult to verify the presence of skin tissues. Perhaps the authors could include some accompanying light images of the sectioned skin tissues?

>> We thank the reviewer for this suggestion. We stained the skin tissues with a Hoechst stain and added these pictures to the supplementary Figure 1.

Supplementary Figure 4. Consider revising the title of the figure so that it is apparent to which experiment these values correspond.

>> The Figure (new number 5) legend is revised. We added the experiment ‘transcriptome sequencing’ (Supplementary Information file, Line 47).

Supplementary Table 9: The table legend should include details about what “-” indicates (not tested? not detected?) as this symbol appears in multiple instances in the table.

>> The table legend of Supplementary Table 9 (new number) is adjusted“-” is changed to NA. In the table legend ‘NA= data not available’ is added (Supplementary information file, line 212).

Abstract and Conclusion Sections: The authors suggest that selective breeding to reduce the presence of galactose carbohydrates in the mucosome is a viable strategy to increase disease resistance and thus conserve susceptible amphibian species. However, the authors should discuss the potential limitations of this approach as well – to my knowledge, the contribution of glycosylation modifications to the function of amphibian epidermal mucus in pathogen defense and the role these carbohydrates play in selection of commensal bacteria (that may produce antifungal metabolites) are largely unexplored. In addition, the authors should be cautious in making this statement given that they acknowledge that presence of galactose is not an accurate predictor of “disease course after infection”

>> We agree with this comment. In the abstract (Line 33), we added “to select for more colonization resistant host lineages”.

In the main text (Line 282-283) we added: “and do not incur harmful side effects such as decreasing the defensive capacity of the skin microbiome.” While we agree with the reviewer that this is certainly plausible, we would like to render this hypothetically (but indeed this will have to be taken into account if this is explored further).

Reviewer #2 (Remarks to the Author):

Major:

1. Line 299. Please indicate that RIPA buffer is Radioimmunoprecipitation assay buffer. Was it made in house or a commercial preparation?

>>Radioimmunoprecipitation assay (RIPA) buffer is added in Line 317. RIPA buffer was purchased from Sigma Aldrich, we corrected this in Line 318.

2. Line 314. Please define the coating buffer more precisely. Is it 3.7 g Sodium Bicarbonate (NaHCO₃) and 0.64 g Sodium Carbonate (Na₂CO₃) per 1 L of distilled water?

>> We defined the coating buffer more precisely on line 333-334. Indeed, the coating buffer is composed of 3.7 g Sodium Bicarbonate (NaHCO₃) and 0.64 g Sodium Carbonate (Na₂CO₃) per 1 L of water.

3. Lines 325-328. Please indicate whether counts of attached zoospores were done in a blinded fashion.

>> The attached zoospores were counted in a blinded fashion. This is added in Line 348.

4. Lines 342-344. Please indicate whether the counts of attached zoospores were done in a blinded fashion.

>> The attached zoospores were counted in a blinded fashion. This is added in Line 365.

5. Lines 346-370. I have read the description of this chemotaxis assay in the original paper (Van Rooij et al. 2015) and again here, and I still can't quite understand it. Is the open end of the capillary in the 400 µl of zoospore solution? Does the closed end stick up and out of the solution at an angle? Since this is a critical assay, it is important to understand. Thus maybe a small schematic diagram of this assay in the supplement would be very helpful for anyone who might like to repeat this assay.

>> A schematic overview of the chemotaxis assay is depicted in supplementary Figure 4.

6. Lines 378-380. Please comment somewhere in the methods or discussion. This is a very short zoospore treatment period. Did you look at any other time points? Were the zoospores newly released? How long after release from the zoosporangium before they settle and form a germ tube? How much of gene expression could be due to "encystment" preparation than invasion or expression of other virulence factors. How clean were the zoospores (no more advanced thalli present)?

>> We used freshly released spores. Release was induced by replacing culture medium by water in a mature culture and spores were used within 2 hours of release when spores were fully motile. We indeed kept treatment periods as short as possible to avoid settlement and encystment, yet such short treatment periods allow quantifying gene expression patterns. We clarified the methods in this section (Lines 401-402).

7. Line 445. Please add the temperature of incubation. I assume it was 15°C.

>> We added the temperature of incubation at Line 476.

8. Lines 482-490. Please briefly describe how the mucosome samples were collected and processed.

>> The collection of the mucosomes is added at Line 514.

9. Lines 541-542. Please indicate the volume of water per area of skin for mucosome collection.

>> The formula to calculate the volume of water per surface area is added on Line 574-577.

Minor:

1. Line 316. I suggest this minor change to the wording "One hundred µl aliquots...".

>> Hundred changed to One hundred in Line 336.

2. Line 322. Change to “One hundred”.

>> Hundred changed to One hundred in Line 342.

3. Line 505. Please indicate the species name (*communis*) of *Ricinus communis* in lower case and italics.

>> Corrected on Line 538.

4. Line 352. Add the word “with”, i.e., "filled with 60 µl carbohydrate".

>> Added on Line 379.

5. Line 422. Please define RPKM as Reads per kilobase of transcript per million mapped reads.

>> Added on Line 452-453.

6. Line 561. Replace the word "normally" with "normal".

>> Corrected.

7. Line 567. Define the RCA scoring here again. Remind the readers at this point of the nature of the lectin, *Ricinus communis* agglutinin.

>> The RCA scoring is added to Lines 615-616.

Reviewer #3 (Remarks to the Author):

My only concern about the manuscript is that, as presented, it's very difficult to assess how robust the modelling of the data are. Throughout it's clear that there are both biological and technical replicates, which the authors do explain clearly. But I worry that these technical replicates are being fed into models and treated as independent degrees of freedom.

>> Please see our reply to a similar comment by the first reviewer. We have redone all analyses using (generalized) linear mixed models, using a nested random effect to take into account the biological and technical replicates.

For example, in your skin lysate binding assay (L291-), 10 newts have their sloughed skin collected. You perform 3 independent experiments with 3 technical replicates per experiment. But each of these 9 data points can only ever come from 1 newt's sample. So they are all technical replicates and should be treated as such. Related to this, if feeding these individual data points into models, t tests (Fig 3) and Kruskal-Wallis tests (Fig 1) become inappropriate.

>> Please see the comments above. Analyses have been redone, and Kruskal-Wallis tests are no longer used.

In addition, lots of the modelling for multi-species experiments seems to pool data from multiple species in linear modelling frameworks, so not in fact accounting for the fact that residuals are not exchangeable between species. Collectively these approaches, if I've understood correctly, will overinflate perceived statistical power. Some clarification here will help because if I've misunderstood then others will too.

>> For Figure 5 and 6, data from multiple species are indeed used, but data do not represent measurements on the same individuals as data are pooled from several sources, including literature review of published papers. Therefore, data have first been summarized at the species-level, and in the analyses, each data point thus represents an independent sample.

We have explored the suggestion of the reviewer to the extent possible (see below), by running models that include species identity as a random effect. We however find very similar results and hence propose to keep the figures and results as presented in the main ms.

Figure 5a: Relationship between peak infection loads and RCA staining. For 14 species (representing 145 individuals), peak infection loads per individual were available. We assigned the species-level RCA values to each observation, and then ran a GLMM to test for the relationship between peak infection loads and RCA staining, specifying 'species' as a random effect. As both overdispersion and zero-inflation were present in the data, models were fit using the glmmTMB function or R package 'glmmTMB', using a negative binomial distribution and zero-inflation correction. We found a strong relationship: RCA estimate and se 1.77 ± 0.71 , z-value = 2.485, $P = 0.013$. The marginal R-square of this model (representing the contribution of the fixed effects (i.e. RCA in this case) is 0.22, the conditional R² including the contribution on the random effects (i.e. species in this case) is 0.50.

Figure 5b: Relationship between mortality and RCA staining. As data on individual mortality were available, as above, we assigned the species-level RCA values to each observation. We then used a binomial linear mixed model (0:1, dead or not) to test for the relationship between probability of mortality and RCA staining, specifying 'species' as a random effect. We again found a strong effect: RCA estimate and se 4.13 ± 1.67 , z-value = 2.473, $P = 0.013$. The marginal Nagelkerke R² of this binomial model is 0.55, the conditional R² is 0.91.

Figure 6a: Relationship between peak infection loads and the percentage of free galactose. By assigning species-level measurements of free galactose to the peak infection load data (see above Fig. 5a), we could run a linear mixed model to test for the relationship between peak infection loads and free galactose, specifying 'species' as a random effect. As both overdispersion and zero-inflation were present in the data, models were fit using the glmmTMB function or R package 'glmmTMB', using a negative binomial distribution and zero-inflation correction. A quadratic relation provided the best fit (quadratic term estimate and se: -0.63 ± 0.22 , z-value = -2.846, $P = 0.0044$). The marginal R² of this model is 0.37, the conditional R² is 0.57.

Figure 6b: Relationship between mortality and the percentage of free galactose. By assigning species-level measurements of free galactose to the mortality data (see above Fig. 5a), we could run a binomial linear mixed model (0:1; dead or not) to test for the relationship between peak infection loads and free galactose, specifying 'species' as a random effect. A quadratic relation provided the best fit (quadratic term estimate and se: -1.18 ± 0.57 , z-value = -2.051, $P = 0.04$). The marginal R² of this model is 0.64, the conditional R² is 0.90.

Figure 6c: Relationship between RCA score and the percentage of free galactose. Sufficient individual measurement data are not available and only a species-level model is possible (as presented in the ms).

Line 565: missing version number for glmmTMB. Also missing citations for all of these packages

>> We have added the missing version numbers and citations 55-59.

Reviewers' Comments:

Reviewer #1:

Remarks to the Author:

The authors have addressed all of my previous comments effectively and I believe the manuscript is now acceptable for publication with the following couple typographical/style changes outlined below (resulting from the production of new graphs):

1. The "2" in "H2O" should be in subscript in Fig. 1C and Fig. 3.
2. Fig. 1 and Fig. 3 are stylistically different from the rest of the figures. Perhaps the authors could adopt a style a bit more consistent with the other figures (and previous versions of the figures)? For example, showing individual data points for biological replicates, having solid lines for the axis, and using an uppercase letter at the beginning of the axis titles. I believe there is a way to change the appearance of figures generated in R. However, I leave this to the discretion of the authors/editors.

Reviewer #3:

Remarks to the Author:

The authors have addressed my comments from the first round of review, and I'm satisfied that the data are being modelled appropriately.

My only remaining point for consideration is the non-linear relationships presented between free galactose and Bd load and mortality, respectively. It isn't really discussed that as per the model predictions having 'too much' free galactose can actually reduce your mortality risk. But this is almost certainly function of variation among species (not all species show the same reaction norm) rather than a general relationship (any species could reduce risk of mortality by increasing free galactose). You tackle this briefly by talking about 'exceptions' and pointing out some of the larger positive and negative residuals, but it might be worth making this among-species heterogeneity point more clear in the manuscript

A wonderfully comprehensive paper with important findings.

Reviewer 1

The authors have addressed all of my previous comments effectively and I believe the manuscript is now acceptable for publication with the following couple typographical/style changes outlined below (resulting from the production of new graphs):

1. The "2" in "H₂O" should be in subscript in Fig. 1C and Fig. 3.

>> *Thank you for your remark. We adapted this in the figures.*

2. Fig. 1 and Fig. 3 are stylistically different from the rest of the figures. Perhaps the authors could adopt a style a bit more consistent with the other figures (and previous versions of the figures)? For example, showing individual data points for biological replicates, having solid lines for the axis, and using an uppercase letter at the beginning of the axis titles. I believe there is a way to change the appearance of figures generated in R. However, I leave this to the discretion of the authors/editors.

>> *We tried to make the style of the figures more consistent.*

Reviewer 3

The authors have addressed my comments from the first round of review, and I'm satisfied that the data are being modelled appropriately.

My only remaining point for consideration is the non-linear relationships presented between free galactose and Bd load and mortality, respectively. It isn't really discussed that as per the model predictions having 'too much' free galactose can actually reduce your mortality risk. But this is almost certainly function of variation among species (not all species show the same reaction norm) rather than a general relationship (any species could reduce risk of mortality by increasing free galactose). You tackle this briefly by talking about 'exceptions' and pointing out some of the larger positive and negative residuals, but it might be worth making this among-species heterogeneity point more clear in the manuscript

>> *We tried to tackle this by rephrasing the paragraph. The new paragraph reads as: The proportion of galactose in the total carbohydrate fraction of the skin washes yielded results in line with those obtained from the RCA staining of tissues. The four anuran species, reported to be *B. salamandrivorans* resistant or tolerant, showed a low percentage of free galactose in the total carbohydrate fraction (Fig. 6, Supplementary Fig. 2, Supplementary Table 9). Moderate correlations were observed between the percentage of free galactose with *B. salamandrivorans* infection peak loads (Pearson $r = 0.641$, $p = 0.003$; Fig. 6a) and with mortality rates (Pearson $r = 0.523$, $p = 0.026$; Fig. 6b), though the sensitivity of free galactose likely varies across species, as suggested by *Lyciasalamandra helverseni*, *Salamandra salamandra* and *Calotriton asper* (Fig. 6a, b), which seem better able to tolerate higher free galactose levels than expected based on the observed linear correlations (i.e. observed infection peak loads and mortality rates outside the 95% CI for both Pearson's correlations). Quadratic regression models show that 70.4% of the variance in infection peak loads ($R^2 = 0.704$; Fig. 6a) and 54.3% of the variance in mortality rates ($R^2 = 0.543$; Fig. 6b) can be explained by the percentage of free galactose from the mucosome washes. (Lines 266-280)*